# Peroxisomal lactate dehydrogenase is generated by translational readthrough in mammals

Fabian Schueren[1][†], Thomas Lingner[2][†], Rosemol George[1][†], Julia Hofhuis[1], Corinna Dickel[1], Jutta Gärtner[1]*, Sven Thoms[1]*

[1]Department of Pediatrics and Adolescent Medicine, University Medical Center, Georg-August-University Göttingen, Göttingen, Germany; [2]Department of Bioinformatics, Institute for Microbiology and Genetics, Georg-August-University Göttingen, Göttingen, Germany

**Abstract** Translational readthrough gives rise to low abundance proteins with C-terminal extensions beyond the stop codon. To identify functional translational readthrough, we estimated the readthrough propensity (RTP) of all stop codon contexts of the human genome by a new regression model in silico, identified a nucleotide consensus motif for high RTP by using this model, and analyzed all readthrough extensions in silico with a new predictor for peroxisomal targeting signal type 1 (PTS1). Lactate dehydrogenase B (LDHB) showed the highest combined RTP and PTS1 probability. Experimentally we show that at least 1.6% of the total cellular LDHB is targeted to the peroxisome by a conserved hidden PTS1. The readthrough-extended lactate dehydrogenase subunit LDHBx can also co-import LDHA, the other LDH subunit, into peroxisomes. Peroxisomal LDH is conserved in mammals and likely contributes to redox equivalent regeneration in peroxisomes.

*For correspondence: gaertnj@ med.uni-goettingen.de (JG); sven.thoms@med.uni-goettingen. de (ST)

[†]These authors contributed equally to this work

**Competing interests:** The authors declare that no competing interests exist.

**Reviewing editor**: Nahum Sonenberg, McGill University, Canada

## Introduction

Translation of genetic information encoded in mRNAs into proteins is carried out by ribosomes. When a stop codon enters the ribosomal A site, release factors bind the stop codon, hydrolyze the peptidyl-tRNA bond, and trigger the release of the polypeptide from the ribosome. If instead of release factor 1 (eRF1), a near-cognate aminoacyl-tRNA pairs with the stop codon in the ribosomal A site, the stop signal is suppressed. Such decoding of a stop codon as a sense codon is known as translational readthrough. As a consequence, translation continues to the next stop codon resulting in the synthesis of C-terminally extended proteins (*Baranov et al., 2002*; *Namy et al., 2004*; *Firth and Brierley, 2012*). Mutant tRNAs are classic stop codon suppressors, but termination also occurs with less than 100% efficiency in normal physiology.

A number of cis-elements on the mRNA, typically 3′ of the stop codon together with trans-acting factors, are known to influence stop codon readthrough (*Firth et al., 2011*). A case of translational readthrough dependent on RNA cis-elements has recently been found and termed programmed translational readthrough (PTR) (*Eswarappa et al., 2014*). But it is also known that the stop codon itself and the nucleotides before and after the stop codon affect readthrough. The three stop codons differ in their tendency to be suppressed. In human, UAA is least and UGA is most likely to allow readthrough (*Beier and Grimm, 2001*; *Baranov et al., 2002*). Studies also show that the nucleotide immediately downstream of the stop codon is biased and can strongly influence readthrough (*McCaughan et al., 1995*). We here define translational readthrough that is entirely dependent on the stop codon and the nucleotides in its immediate vicinity as basal translational readthrough (BTR). Thus BTR is independent of cis-acting elements and also differs from pharmacologically induced

**eLife digest** Amino acids are the building blocks of proteins, and the order of the amino acids in a protein is determined by the order in which 'codons' appear in a messenger RNA molecule. Most codons represent a specific amino acid, but there are also three stop codons that are used to mark the end of a protein.

When the cellular machinery that 'translates' the messenger RNA molecule into a protein encounters a stop codon, it stops and releases the completed protein. Sometimes, however, the stop codon is not interpreted as a stop signal, and the translation of the messenger RNA molecule continues until another stop codon is encountered. This process is known as readthrough.

Some organisms, in particular viruses and fungi, use readthrough to produce a wider range of proteins than their genomes would otherwise allow. While readthrough also occurs in higher organisms such as mammals, it is not known if the resulting proteins perform extra functions that the original protein does not perform.

A number of factors affect whether readthrough occurs when an mRNA template is being translated. For example, each of the three stop codons has a different likelihood of having its stop signal misinterpreted, and the mRNA sequence that surrounds the stop codon can also affect the likelihood of readthrough.

Schueren et al. have developed a computational model that estimates how common this form of translational readthrough is in the human genome. The model was based on the identity of the stop codons themselves and the surrounding mRNA sequence. This model was then combined with another model that identifies proteins that are targeted to a structure inside a cell called the peroxisome, which is where a number of essential energy-releasing reactions take place. The combined model enabled Schueren et al. to identify proteins that both perform functions in the peroxisome and are likely to be formed by readthrough.

The combined model suggested a protein that is a part of lactate dehydrogenase: an enzyme that speeds up chemical reactions that are important for the cell to produce energy. Low levels of lactate dehydrogenase had previously been found in the peroxisome, despite it apparently lacking a specific sequence of amino acids that proteins need to have to enter the peroxisome. However, Schueren et al. confirmed experimentally that readthrough does occur for the lactate dehydrogenase component identified by the model, revealing that it contains a 'hidden' peroxisome-targeting region. Furthermore, when more translational readthrough occurred, more lactate dehydrogenase was found in the peroxisomes.

This unusual way that lactate dehydrogenase enters the peroxisome is an example of how the cell optimizes the used of the genetic information encoded in the genome and in messenger RNA. Translational readthrough always ensures that a certain proportion of lactate dehydrogenase will be brought to the peroxisome. The computational model developed here will be a valuable tool to identify other such proteins produced from genomes, including the human genome and those of other species.

readthrough. Induction of readthrough, most prominently by aminoglycoside antibiotics, is an attractive strategy in the treatment of the large number of genetic disorders caused by premature stop codons (*Bidou et al., 2012*; *Keeling et al., 2014*).

In viruses, readthrough optimizes the coding capacity of compact genomes (*Firth and Brierley, 2012*). In the yeast *Saccharomyces cerevisiae*, the eukaryotic release factor eRF3 can form prion-like polymers, which introduces a level of epigenetic regulation not found in other eukaryotes (*Tuite and Cox, 2003*). In fungi, translational readthrough extends cytosolic glycolytic enzymes by a cryptic peroxisomal targeting signal (*Freitag et al., 2012*). In *Drosophila*, readthrough is known to affect between 200 and 300 proteins (*Jungreis et al., 2011*; *Dunn et al., 2013*), and in mammals readthrough has been described for more than 50 individual transcripts (*Geller and Rich, 1980*; *Chittum et al., 1998*; *Yamaguchi et al., 2012*; *Dunn et al., 2013*; *Eswarappa et al., 2014*; *Loughran et al., 2014*). Ribosome profiling and phylogenetic approaches provide powerful methods for the systematic identification of readthrough in mammals (*Jungreis et al., 2011*; *Dunn et al., 2013*; *Eswarappa et al., 2014*; *Loughran et al., 2014*).

We wanted to find a physiological role for translational readthrough in humans by identifying C-terminal extensions with targeting signals that would create a functional difference between the normal and the readthrough-extended form. To achieve this aim, we concentrated on proteins deriving from BTR. Based on experimental data, we assigned regression coefficients to all possible nucleotides in the stop codon context (SCC) and, using those regression coefficients, estimated the readthrough propensity (RTP) of all stop codons in the human genome or transcriptome. We were able to formally derive a new nucleotide consensus for high RTP from the regression coefficients of our model. Then we screened all predicted C-terminal extensions for peroxisomal targeting signals because peroxisomes import most of their matrix proteins through a short targeting signal (PTS1) at the very C-terminus (*Smith and Aitchison, 2013*). We here show that lactate dehydrogenase B (LDHB) combines a very high translational readthrough with a hidden, yet functional and evolutionarily conserved, PTS1. This peroxisomal isoform of LDH, containing the readthrough-extended LDHBx subunit, is likely to be involved in the regeneration of redox equivalents for peroxisomal β-oxidation.

## Results

### Genome-wide in silico analysis of basal translational readthrough

In order to develop a computational method to assess the RTP of all human SCCs that would allow the identification of genes with high BTR, we focused on SCCs comprising 15 nucleotides including and surrounding the stop codon (nucleotides −6 to +9, stop codon at positions 1 to 3). In order to calculate linear regression between the SCCs and their experimental BTR values, we formalized SCCs using a binary vector that represented the stop context in a multi-dimensional vector space (*Figure 1A* and *Figure 1—figure supplement 1*). The three stop codons were condensed into one position, so that the binary vector required 51 dimensions, for the four possible nucleotides in the six positions before and after the stop codon, and for the three stop codons (12 × 4 + 3). The vector was combined with experimentally accessible BTR frequencies. For the first approximation model (LIN), we used 66 sequences derived from human nonsense mutations (*Floquet et al., 2012*). The nucleotide sequences of these stop contexts show no bias with respect to RTP, because the contexts and the stop codons evolved independently, and therefore the context nucleotides are random in relation to the stop codon. We calculated a linear regression model for these SCCs and used only the experimental BTR values that had been measured in the absence of aminoglycosides. The model assigns regression coefficients to all possible nucleotides in the stop context (*Figure 1—figure supplement 1*).

For a first round of whole-genome RTP prediction, we extracted the SCCs for each transcript from the Ensembl database and calculated RTP by adding up the regression coefficients of all relevant positions. An outline of this algorithm is shown in *Figure 1A* and in more detail in *Figure 1—figure supplement 1*. A sortable list of LIN RTP values for all human transcripts is contained in Dataset 1 (*Schueren et al., 2014*).

To expand the data basis of the RTP algorithm and to obtain evidence that the algorithm indeed predicts BTR values, we selected candidate transcripts with high, intermediate, and low RTP and tested them using a dual reporter assay (*Figure 1B* and *Table 1*). For experimental analysis, SCCs spanning 10 nucleotides upstream and downstream of the stop codon were expressed with a 5′/N-terminal yellow fluorescent protein (Venus) and a 3′/C-terminal humanized Renilla luciferase (hRluc) tag. Stop suppression leads to the expression of hRluc, and Venus served as an internal expression control. Readthrough is expressed as luciferase activity per Venus fluorescence. This approach excludes introns and exon junction complexes and, due to the relatively short stretch of variable nucleotides between the reporters, also does not allow for extensive RNA structures that could modulate readthrough. Consequently, this form of the dual reporter assay focuses on the assessment of BTR not influenced by specific cis-elements. The additional candidates tested showed BTR between 0.10% (±0.006%) and 2.91% (±0.15%) relative to the 100% readthrough control expressing the Venus-hRluc fusion protein without an intervening stop codon region (*Figure 1C* and *Table 1*). The aminoglycoside antibiotic geneticin (G418) increased readthrough by between 3.25 (±0.41) and 40.38 (±5.33)-fold (*Figure 1C*). Geneticin could only increase the luciferase-per-Venus signal when a stop codon separated Venus and luciferase, indicating that our dual reporter assay faithfully reports readthrough. The finding that experimental readthrough could be increased by treatment with aminoglycosides also excludes alternative mechanisms such as RNA editing or splicing that might explain the relative increase of the luciferase over the Venus signal. The highest levels of induction can only be reached when

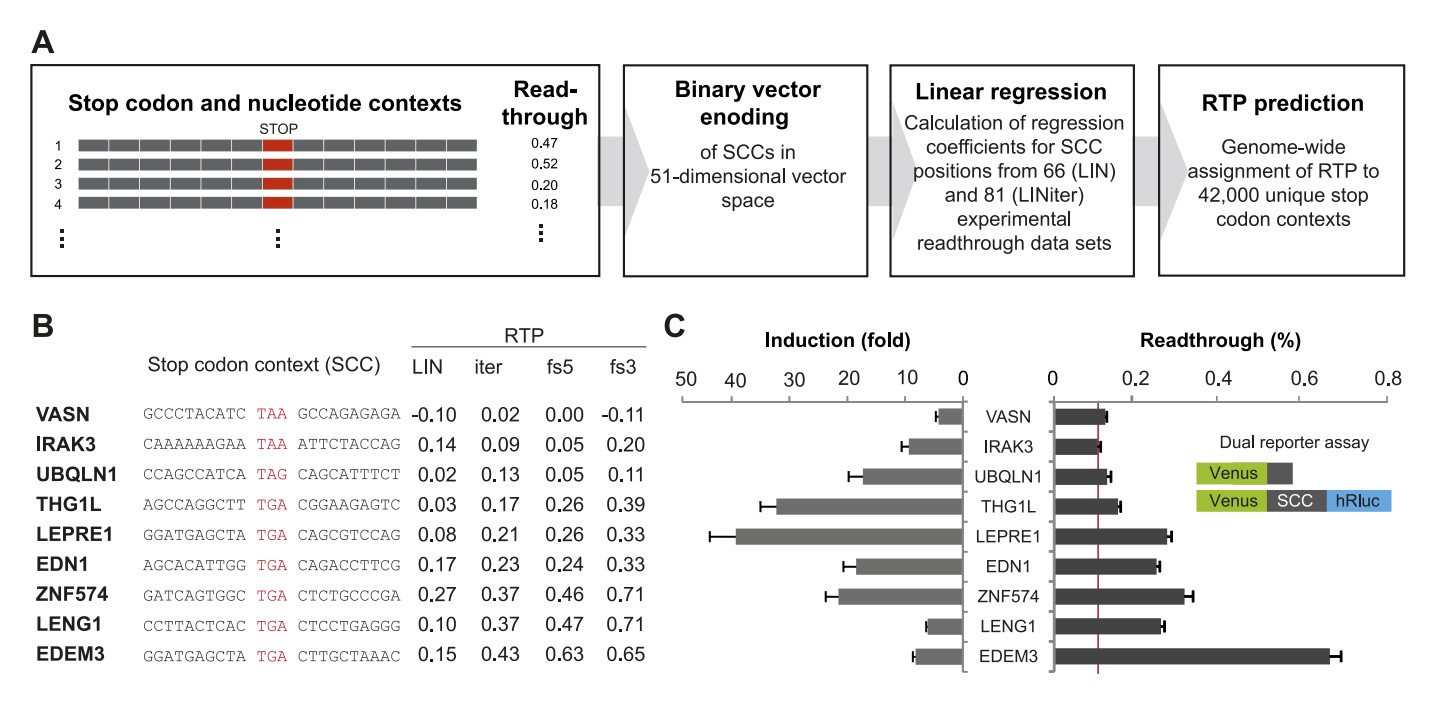

**Figure 1**. Genome-wide in silico analysis of basal translational readthrough (BTR) in humans. (**A**) Schematic representation of the readthrough propensity (RTP) predictor algorithm. Binary vector representations of stop codon contexts and their experimental readthrough values are used to determine the coefficients of a linear regression model. For prediction of RTP for a given stop codon context, the position-specific regression coefficients are added up. (**B**) RTP for selected human transcripts. LIN denotes first-pass RTP calculations, LINiter iterative improvement of RTP scoring, and LINfs3 and LINfs5 the reduced models. The RTP of all human transcripts can be found in Dataset 1 (*Schueren et al., 2014*). (**C**) Experimental readthrough by dual reporter assay in HeLa cells. Readthrough is expressed as luciferase per Venus signal. The red line marks the background readthrough level obtained from a construct containing two contiguous UAA stop codons separating the Venus and the hRluc. The aminoglycoside geneticin (100 μg/ml) induces translational readthrough. SCC: stop codon context; hRluc: humanized Renilla luciferase. Error bars, SD.

The following figure supplements are available for figure 1:

**Figure supplement 1**. Schematic representation of the readthrough propensity (RTP) prediction procedure.

**Figure supplement 2**. Correlation of RTP and BTR in the LINiter model.

basal readthrough is low, and, vice versa, a high BTR somewhat limits the maximum induction factor (*Figure 1C*), suggesting that maximal BTR readthrough is limited to levels below 15%.

Next we added our candidate sequences and their experimentally determined readthrough levels to obtain an iterative and extended RTP model (LINiter). Again, we applied this model to all human transcripts (see *Schueren et al., 2014* for Dataset 1; model parameters are shown in *Table 2*). We measured the correlation of RTP and experimental BTR by leave-one-out cross-validation during computation of the regression coefficients. For the LINiter model, we obtained a weak but significant Pearson correlation coefficient of 0.34 (p = 0.002) (*Figure 1—figure supplement 2*). To determine the origin of the apparently strong non-linear contribution to RTP, we analyzed the regression coefficients of the LINiter model. Nucleotide positions associated with coefficients of large absolute value contribute most to RTP. The relative contribution of nucleotides within the SCC to the readthrough prediction is shown in *Figure 2A*.

## A consensus for high readthrough in humans

The sequence-logo representation of regression factors in *Figure 2A* indicates that the three or four nucleotides following the stop codon contribute to readthrough. The quantitative manner in which we derived LINiter values allowed us to rationally derive a nucleotide motif permitting high readthrough in humans. We identified the nucleotide positions with the strongest influence on BTR in humans by

**Table 1.** Additional experimental dual reporter readthrough data of stop codon context constructs used for the LINiter model

| Gene symbol | Stop codon context | Readthrough (%) (SD) |
|---|---|---|
| ZNF-574 | GATCAGTGGC TGA CTCTGCCCGA | 0.31 (0.020) |
| LDHB | AAAAGACCTG TGA CTAGTGAGCT | 1.55 (0.087) |
| PPP1R3F | ATTCTCCCAA TAA AGCTTTACAG | 0.18 (0.009) |
| LDHB [TGAT] | AAAAGACCTG TGA TTAGTGAGCT | 0.17 (0.009) |
| LDHB [TAA] | AAAAGACCTG TAA CTAGTGAGCT | 0.20 (0.009) |
| LDHB [TAAT] | AAAAGACCTG TAA TTAGTGAGCT | 0.17 (0.009) |
| LENG1 | CCTTACTCAC TGA CTCCTGAGGG | 0.26 (0.009) |
| VASN | GCCCTACATC TAA GCCAGAGAGA | 0.12 (0.004) |
| MDH1 | TTCCTCTGCC TGA CTAGACAATG | 2.91 (0.147) |
| PRDM10 | CACCAAACCA TGA CTTCCACCCT | 0.13 (0.005) |
| FBXL20 | CATCATCCTA TGA CAATGGAGGT | 0.10 (0.006) |
| THG1L | AGCCAGGCTT TGA CGGAAGAGTC | 0.15 (0.006) |
| EDEM3 | GGATGAGCTA TGA CTTGCTAAAC | 0.66 (0.027) |
| EDN1 | AGCACATTGG TGA CAGACCTTCG | 0.25 (0.008) |
| UBQLN1 | CCAGCCATCA TAG CAGCATTTCT | 0.13 (0.009) |
| IRAK3 | CAAAAAAGAA TAA ATTCTACCAG | 0.10 (0.007) |
| SLC3A1 | TACCTCGTGT TAG GCACCTTTAT | 0.18 (0.008) |
| LEPRE1 | GGATGAGCTA TGA CAGCGTCCAG | 0.27 (0.010) |

Stop codon constructs expressing plus/minus 10 nucleotides were analyzed in HeLa cells.

feature selection, that is by successively eliminating those positions that contribute least to the prediction (*Figure 2B*). One by one the nucleotide positions with the smallest sum of squared regression coefficients were removed from the model. We find that two reduced models improve the prediction. Models with either five or three relevant context positions in addition to the stop codon correspond to the local and global residual error minimum, respectively. LINfs5 comprises nucleotide position −6, the stop codon, and positions +4 to +7, and LINfs3 comprises only the stop codon and positions +4 to +6, that is the codon following the stop (*Figure 2B*). The results of this analysis indicate that in humans the stop codon and the three nucleotides immediately downstream of the stop codon have the largest influence on BTR (LINfs3). The corresponding consensus is <u>UGA</u> CUA (stop codon underlined). Possibly also the nucleotides at positions +7 (the fourth position after the stop) and −6 contribute to BTR. The RTP-BTR correlation associated with LINfs3 was 0.41 (p = 0.0001) (*Figure 2—figure supplement 1*). To test if the LINfs3 consensus indeed confers high BTR, we analyzed four additional candidate SCCs. Three high-RTP SCCs were derived from AQP4, SYTL2, and CACNA2D4, and DHX38 was used as a control with a low RTP. AQP4, SYTL2, and CACNA2D4 conform with the LINfs3 consensus, whereas DHX38 does not. AQP4, SYTL2, and CACNA2D4 showed 2.29% (±0.09%), 0.99% (±0.06%), and 0.61% (±0.02%) readthrough in HeLa cells, whereas for DHX38 readthrough was only 0.27% (±0.04%) (*Figure 2C*), confirming that LINfs3 SCC indeed allows a very high rate of stop suppression. Next we wanted to test if these conclusions obtained in HeLa cells can be extended to other cell types. We therefore performed dual reporter experiments using the HT1080 fibrosarcoma cell line, the human embryonic kidney cell line (HEK), and the U373 cell line. In all these experiments, the relative distribution of BTR values remained the same, with AQP4 showing the highest and DHX38 the lowest BTR (*Figure 2C*). The finding that readthrough is lower in CACNA2D4 than in AQP4 and SYTL2 can also be taken as evidence that SCC position +7 (fourth after the stop) makes a contribution. Taken together, these experiments show that BTR is indeed a property of the respective SCC, and that readthrough may be differently regulated in different tissues.

The linear approximation underlying the LINiter and the LINfs3 models led to the identification of the UGA CUA (LINfs3) consensus conferring high BTR. A partially overlapping set of genes with this consensus

**Table 2.** Regression factors of the LINiter and LINfs3 models

**LINiter model (stop codon context position −6 to +9)**

| Base/position | −6 | −5 | −4 | −3 | −2 | −1 | 4 |
|---|---|---|---|---|---|---|---|
| A | −0.00041 | 0.00130 | −0.00028 | −0.00073 | −0.00071 | 0.00016 | −0.00037 |
| C | −0.00105 | 0.00164 | 0.00075 | −0.00004 | 0.00133 | 0.00109 | 0.00375 |
| G | 0.00060 | −0.00077 | −0.00041 | 0.00193 | −0.00048 | 0.00043 | −0.00156 |
| U/T | 0.00200 | −0.00103 | 0.00108 | −0.00002 | 0.00100 | −0.00054 | −0.00067 |

| Base/position | 5 | 6 | 7 | 8 | 9 | Stop | |
|---|---|---|---|---|---|---|---|
| A | −0.00068 | 0.00276 | −0.00020 | 0.00105 | −0.00081 | −0.00026 | TAA |
| C | −0.00097 | −0.00026 | −0.00062 | −0.00017 | 0.00148 | −0.00103 | TAG |
| G | −0.00008 | −0.00059 | 0.00245 | −0.00058 | 0.00014 | 0.00243 | TGA |
| U/T | 0.00287 | −0.00076 | −0.00049 | 0.00084 | 0.00032 | | |

**LINfs3 model (Stop and position +4 to +6)**

| Base/position | 4 | 5 | 6 | Stop | |
|---|---|---|---|---|---|
| A | 0.00006 | −0.00071 | 0.00306 | 0.00005 | TAA |
| C | 0.00351 | −0.00056 | 0.00021 | −0.00052 | TAG |
| G | −0.00111 | 0.00010 | −0.00093 | 0.00229 | TGA |
| U/T | −0.00064 | 0.00299 | −0.00053 | | |

These model weights are 'raw', that is as obtained from the ridge regression procedure. For prediction of RTP, the weights associated with nucleotides within the stop codon context and the corresponding stop codon have to be added up. For calculation of our RTP score, we normalized the model weight vectors (i.e., the complete stack of weights) to Euclidean unit sum which corresponds to a division of weights by 0.0088 (LINiter) and 0.0063 (LINfs3), respectively. Furthermore, the sequence feature vectors were normalized to Euclidean unit sum which corresponds to a division by the square root of the length (3.6 and 2, respectively). As a shortcut to this, the sum of raw scores can be divided by 0.0317 and 0.0126, respectively.

was recently tested (*Loughran et al., 2014*). An overview of all experimentally confirmed cases of translational readthrough shown in *Figure 2—figure supplement 2* reveals that ribosome profiling, phylogenetic approaches, and RTP screening are complementary approaches. For example, only one of the 42 readthrough genes found by ribosome profiling in foreskin fibroblasts (*Dunn et al., 2013*) contains the UGA CUA consensus. The widely varying levels and sequence requirements for efficient stop codon suppression suggest that multiple molecular mechanisms can cause readthrough in mammals.

## Identification of peroxisomal targeting signals in readthrough extensions

The genome-wide in silico analysis of RTP provides the basis for the identification of the physiological functions of a readthrough protein. We have therefore screened the extensions for possible elements that could confer functional differences between the normal and the extended form of the protein. We screened the extensions for possible transmembrane domains (*Krogh et al., 2001*), for prenylation sites (*Zhang and Casey, 1996*), for endoplasmic retention signals (*Zerangue et al., 2001*; *Stornaiuolo et al., 2003*), and for glycosylation sites (*Zielinska et al., 2010*; *Schwarz and Aebi, 2011*).

To identify genes with a high BTR and a readthrough extension conferring a biological function, we decided to focus on the detection of proteins carrying a hidden peroxisomal targeting signal type 1 (PTS1) in the extension. This targeting mechanism had been shown to divert a small fraction of cytosolic glycolytic proteins to peroxisomes in fungi (*Freitag et al., 2012*). PTS1 cover more than 90% of the targeting motifs of peroxisomal matrix proteins. The alternative PTS2 is found in only very few matrix proteins, and has even been lost in some organisms (*Lanyon-Hogg et al., 2010*). PTS1 is localized at the very C-terminus of a substrate protein. However, the quintessential PTS1, Ser-Lys-Leu (SKL), is neither necessary nor sufficient to support matrix protein import into peroxisomes. Variations exist, and amino acids upstream of the terminal tripeptide also contribute to targeting (*Brocard and Hartig, 2006*).

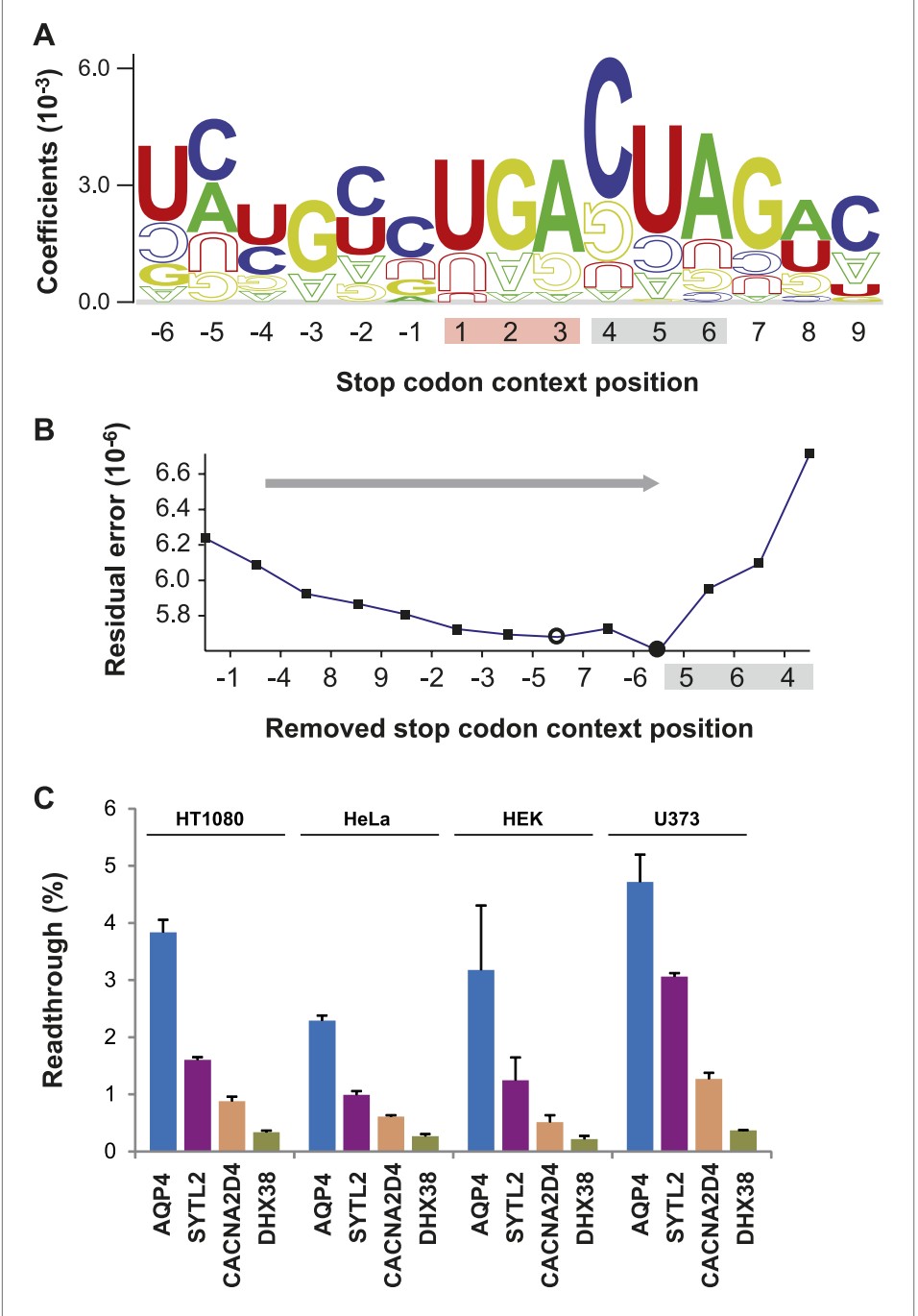

**Figure 2**. Characterization of basal translational readthrough (BTR): consensus and candidates. (**A**) Sequence logo plot of regression coefficients of stop codon contexts (SCCs) in the LINiter model. Character size corresponds to regression coefficients. The model treats stop codons as one nucleotide position. Filled/upside-down letters correspond to positive/negative coefficients, respectively. (**B**) Consensus motif for high readthrough propensity (RTP) derived from the predictive model. The stop codon together with the nucleotide triplet following the stop codon provides the best predictor for RTP. The consensus was derived by feature selection: starting from LINiter, positions with the least contribution to prediction were successively eliminated as indicated by the gray arrow. Nucleotide positions on the x-axis mark the removed positions upon transition to a reduced model. LINfs3 (UGA CUA, stop codon underlined) represents the global minimum of regression error (filled circle). The model LINfs5, corresponding to a local minimum, additionally encompasses positions +7 and −6, indicating that these positions could also contribute to high BTR. (**C**) BTR determination of candidates from the genome-wide in silico screen. Dual reporter assays with Venus and humanized Renilla luciferase containing SCCs from AQP4 (UGA CUA G), SYTL2 (UGA CUA G), CACNA2D4

*Figure 2. Continued on next page*

*Figure 2. Continued*

(UGA CUA T), and DHX38 (UGA CUU G). AQP4, SYTL2, and CACNA2D4 reveal high BTR in all tissues tested. HT1080, human fibrosarcoma cell line; U373, glioblastoma cell line. HEK, human embryonic kidney cells. Error bars, SD.

The following figure supplements are available for figure 2:

**Figure supplement 1**. Correlation of RTP and BTR in the LINfs3 model.

**Figure supplement 2**. Translational readthrough in humans.

Moreover, PTS1 does not confer a binary decision (to import or not to import), but is likely to determine an equilibrium between cytosolic and peroxisomal localization. This is best exemplified by the peroxisomal marker protein catalase, a considerable amount of which is not imported into peroxisomes due to an inherently weak PTS1 which is associated with low affinity to the cytosolic PTS1-receptor PEX5 (*Maynard et al., 2004*). We took advantage of these scalable properties of PTS1 and adapted to human PTS1 a prediction algorithm that we had previously developed for plants (*Lingner et al., 2011*). This machine learning-based method has been shown to accurately predict proteins with canonical and non-canonical PTS1 peptides and provides evidence for peroxisome targeting in terms of a posterior probability (*Lingner et al., 2011*).

To program the human PTS1 prediction algorithm, we conducted orthology searches on 24 known human PTS1 sequences in metazoa using BLAST against protein and EST databases. The resulting dataset and several thousand metazoan sequences without peroxisomal association were used as positive and negative examples in a discriminative machine learning setup. Here, the sequences were represented by binary vectors encoding the presence or absence of up to 15 C-terminal amino acids. Models were trained and validated using regularized least squares classifiers (RLSC) and fivefold cross-validation. A more detailed description of the human PTS1 scoring can be found in the 'Materials and methods' section. We calculated the PTS1 posterior probabilities of all predicted C-terminal readthrough extensions derived from the human transcriptome (see *Schueren et al., 2014* for Dataset 1).

## LDHB is extended by translational readthrough

Based on the assumption that a protein is more likely to target to peroxisomes by a cryptic PTS1 when the RTP and the extension's PTS1 scores are high, we used the product of RTP LINiter scores and PTS1 posterior probabilities as a predictor of functional peroxisomal targeting by a hidden PTS1 in the extension (see *Schueren et al., 2014* for Dataset 1). To avoid negative product scores, we scaled RTP between 0 and 1 before multiplication (now designated RTP$^+$).

We identified LDHB, one of the two human lactate dehydrogenase (LDH) subunits, at the top (position 1 of 42,069 entries) of our sorted list of combined RTP$^+$ and PTS1 scores (see *Schueren et al., 2014* for Dataset 1). The distribution of RTP$^+$ × PTS1 product scores over all human transcripts indicates that other candidates must have considerably lower RTPs and/or targeting efficiencies, because the score drops by 50% over the first 40 of 42,069 transcripts (*Figure 3A*).

To experimentally confirm high BTR, we expressed the human LDHB SCC in the Venus/hRluc dual reporter assay. Readthrough was 1.55% (±0.09%) and mutation of the stop codon and/or the consecutive nucleotide strongly suppressed readthrough (*Figure 3B* and *Figure 3—figure supplement 1*). Treatment with geneticin increased readthrough to 4.38% (±0.42%) (compare with induction factors in *Figure 3C*).

To establish that the full-length protein is extended by stop suppression, LDHB including the extension (designated LDHBx for 'extended') and mutants were expressed with N-terminal HA- and C-terminal Myc-tags and analyzed by Western blotting. Full-length LDHB showed aminoglycoside-inducible readthrough, and the loss of readthrough upon exchange of the stop codon or the nucleotide following the stop codon confirms the special function of the LDHB SCC in stimulating translational readthrough (*Figure 3D*).

## Peroxisomal localization of LDHB depends on translational readthrough

The identification of LDHB as virtually the only human protein with a high combined readthrough and peroxisomal targeting probability is surprising, because a peroxisomal readthrough-extended LDHBx entails at least one new LDH isoform. On the other hand, LDH activity and isoforms inside peroxisomes

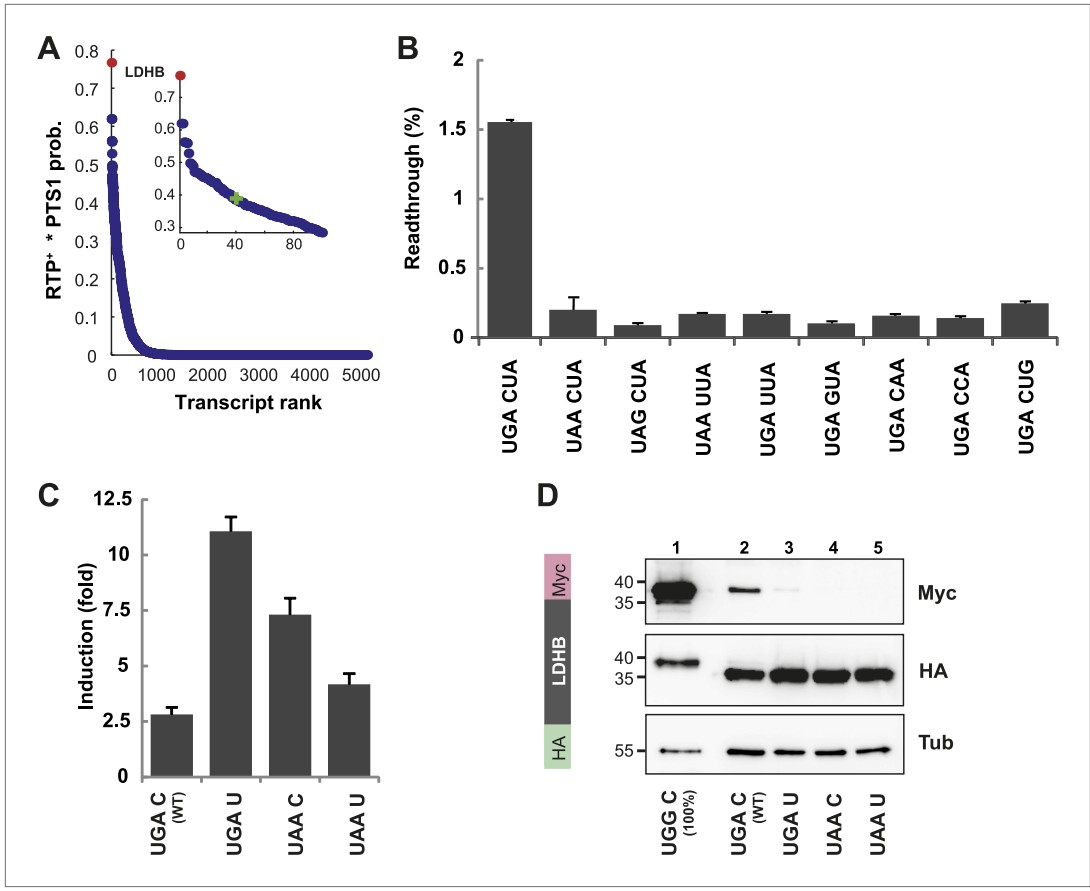

**Figure 3**. LDHB is extended by translational readthrough. (**A**) Genomic distribution of RTP+ × PTS1 product scores. Product scores are 0 for rank 5015 to 42069. Green cross: 50% of maximum score. LDHB has the highest product score, exceeding rank 2 by 24%. RTP+ denotes positively scaled LINiter values. (**B** and **C**) Venus/hRluc dual reporter assay with LDHB wild-type and mutant stop codon contexts. Error bars, SD. (**B**) Wild-type LDHB stop context shows high basal translational readthrough (BTR). Mutational analysis of the LINfs3 consensus of LDHB. Replacement of the stop codon and mutations in positions +4 to +6 reduce readthrough. (**C**) LDHB readthrough induction by the aminoglycoside geneticin. (**D**) Full-length LDHB is extended by readthrough. Western blot of dual tag assay with LDHBx with N-terminal HA- and C-terminal Myc-tag. Molecular mass marker in kDa.

The following figure supplement is available for figure 3:

**Figure supplement 1**. The LDHB stop context favors readthrough (Western blot).

were known for more than four decades (**McGroarty et al., 1974**; **Osmundsen, 1982**; **Völkl and Fahimi, 1985**; **Baumgart et al., 1996**; **McClelland et al., 2003**; **Gronemeyer et al., 2013**). In the apparent absence of known targeting signals, however, it has not been possible to explain how the protein can enter the peroxisome. Therefore we conducted an investigation to determine whether the extended human LDHBx protein and the predicted PTS1 therein lead to peroxisomal localization. We expressed LDHBx as a fusion protein with an N-terminal enhanced yellow fluorescent protein (YFP) and co-labeled cells by immunofluorescence with the peroxisomal marker PEX14, a peroxisomal membrane protein. YFP-LDHB showed the expected cytosolic localization (**Figures 4A** and **5A**). We hypothesized that a large excess of cytosolic YFP-LDHB masks the peroxisomal localization of LDHBx. To remove cytosolic YFP-LDHB, we permeabilized cells by digitonin before fixation and washed out the cytosol using phosphate-buffered saline (PBS). In agreement with peroxisomal targeting through the cryptic PTS1, LDHBx is found localized in peroxisomes after removal of the cytosol (**Figures 4B and 5B**). In control experiments, we show complete removal of cytosolically expressed YFP by cytosol wash-out (**Figure 5—figure supplement 1**) and peroxisomal localization of a YFP variant fused to

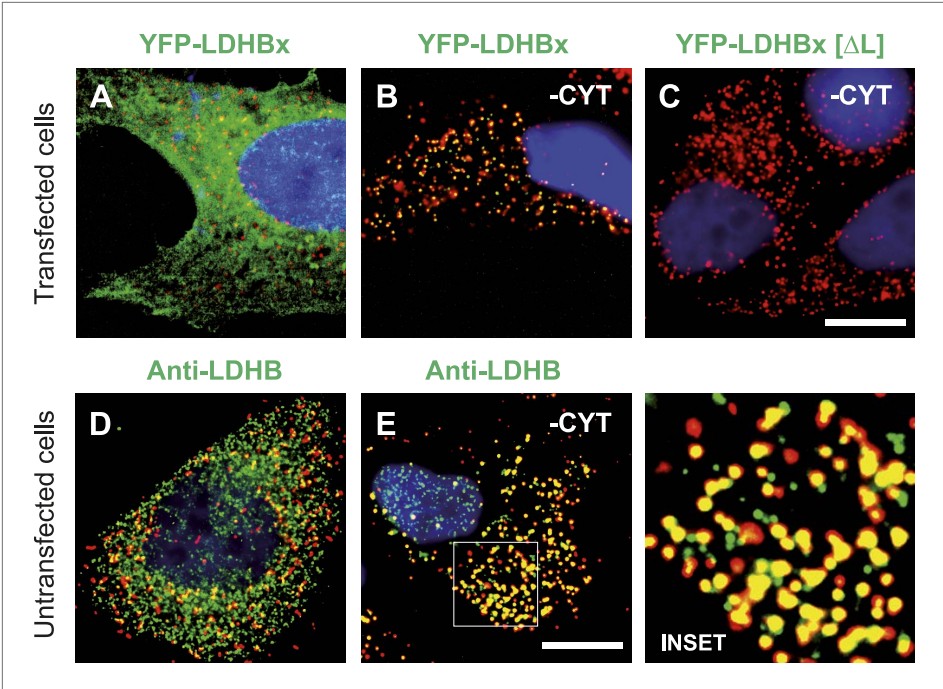

**Figure 4**. LDHBx targets to the peroxisome by translational readthrough and a hidden peroxisomal targeting signal type 1 (PTS1) in the 3' extension. (**A–C**) Direct fluorescence microscopy of transfected HeLa cells. Immunofluorescence with the peroxisome marker anti-PEX14 (red). (**A**) YFP-LDHB (green) mainly localizes to the cytosol. The strong fluorescence signal in the cytosol prevents detection of LDHB in other cellular compartments. (**B**) Upon plasma membrane permeabilization and removal of cytosol (-CYT), a small fraction of LDHB remains co-localized with the peroxisome marker. (**C**) Peroxisomal targeting of LDHB is dependent on the cryptic PTS1 Ser-Arg-Leu (SRL) in the extension. Deletion of the L in SRL blocks import into peroxisomes. (**D** and **E**) Endogenous LDHB is localized to peroxisomes in untransfected wild-type cells. Immunofluorescence with anti-LDHB (green) and anti-PEX14 (red) antibodies. (**D**) Endogenous LDHB is cytosolic. (**E**) Removal of cytosol (-CYT) reveals co-localization with PEX14. Bar 10 μm.

PTS1 of the peroxisomal matrix protein ACOX3 (**Figure 5—figure supplement 2**). To confirm that LDHB targeting to peroxisomes is dependent on the putative PTS1 in the readthrough extension, we changed the SRL terminus (PTS1 probability 94.3%) to SSI (0.002%) and to SR (ΔL, 0.00001%). These mutations blocked YFP-LDHBx targeting to the peroxisome (**Figures 4C and 5C–F**). Remarkably, exchange of the leaky UGA stop with the tighter UAA reduced peroxisomal localization of YFP-LDHB (**Figure 6A,B**). Our results show that the high-RTP SCCs as well as the PTS1 in the extension after the stop codon are needed for peroxisome targeting. The extension must be accessible to ribosomal translation and contain a functional PTS1. It is known that PTS1-dependent targeting guides proteins into peroxisomes and not only to the membrane. The dependence of LDH targeting on the hidden PTS1 and on the nature of the stop codon thus confirms that the protein is indeed inside the peroxisome. As expected, replacing the stop codon by tryptophan-encoding UGG renders LDHBx entirely dependent on the PTS1 (**Figure 6C,D**).

To obtain more direct evidence for the readthrough-dependent low abundance targeting of human LDHB to peroxisomes, we analyzed untransfected wild-type cells by immunofluorescence with anti-LDHB and anti-PEX14 antibodies. LDHB appears distributed in the cytosol (**Figure 4D**). After cytosol depletion, however, the remaining LDHB signal is mainly peroxisomal (**Figure 4E**). A small portion of LDHB may localize to other cellular locations protected against cytosol removal. We confirmed these results in human skin fibroblasts, COS-7 cells (monkey kidney fibroblast line), the human glioblastoma cell line U118, and freshly prepared rat cardiomyocytes (**Figure 7**). Our data are in agreement with readthrough-dependent targeting of about 1.6% of the LDHB to peroxisomes mediated by the cryptic PTS1 in the extension. Remarkably, treatment of untransfected wild-type HeLa cells with geneticin increased LDHBx levels in the peroxisome (induction factor 1.89, n = 28, $t$ test p < 0.0001) suggesting

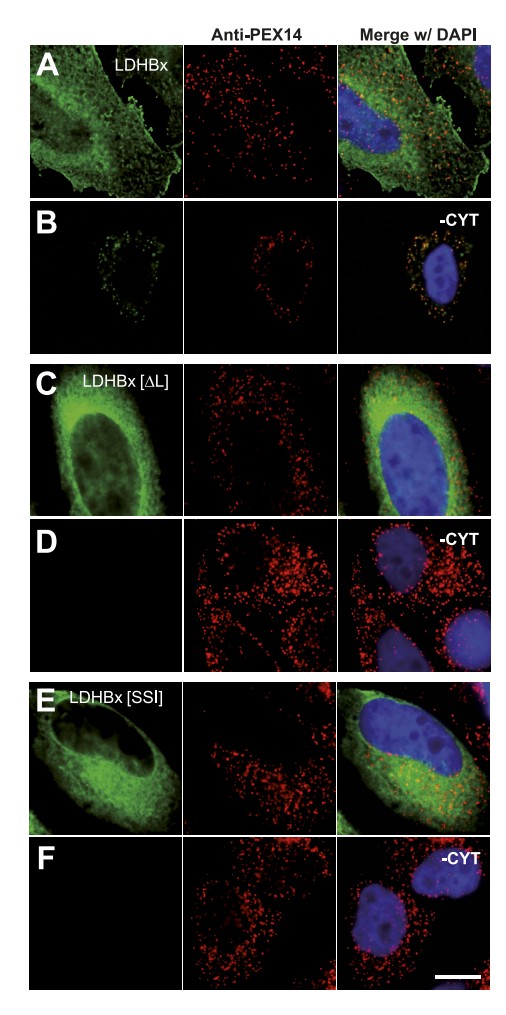

**Figure 5**. Peroxisome targeting of LDHBx is dependent on a hidden peroxisomal targeting signal in the readthrough extension. Combined direct fluorescence and immunofluorescence in HeLa cells. (**A**) YFP-LDHBx expression: LDHBx is mainly cytosolic. (**B**) LDHBx targets to the peroxisome. Cells were permeabilized with digitonin, and cytosol was removed by washing with phosphate-buffered saline. (**C**–**F**) Mutation of the cryptic PTS1 in the extension blocks peroxisomal targeting of LDHBx. (**C** and **D**) Deletion of the amino acid L of the SRL in the PTS1 readthrough extension gives a wild-type cytosolic localization of LDHB and blocks import into the peroxisome completely. (**E** and **F**) Similarly, the SRL-to-SSI substitution does not interfere with cytosolic expression of the LDHB but completely blocks peroxisomal localization of LDHBx[SSI]. Bar 10 μm.

The following figure supplements are available for figure 5:

**Figure supplement 1**. Permeabilization by digitonin allows complete removal of cytosol.

**Figure supplement 2**. Cell permeabilization and removal of cytosol maintains peroxisomal integrity and co-localization of peroxisome marker (positive control).

elevated peroxisomal LDHBx levels as a general pharmacological consequence of aminoglycoside treatment.

Next we wanted to test if there is evidence for differential regulation of translational readthrough of LDHB in different cell types. We expressed LDHB and mutant dual reporter constructs in COS-7 cells, U118 cells, and HEK cells. Readthrough of LDHB ranged between 1.55% (±0.09%) in HEK and HeLa and 1.88% (±0.14%) in COS-7. Surprisingly, in U118 cells LDHB readthrough is increased to 5.09% (±1.03%) (**Figure 8**). Geneticin induced readthrough by factors ranging between 1.32 (±0.09) and 2.82 (±0.27) (**Figure 8**). LDHB stop suppression is thus not restricted to special tissues, and may be differently regulated in different cell types.

Analysis of animal LDHB orthologs in vertebrates shows that PTS1 in the extension is exclusively and strictly conserved in mammals, supporting the notion of a functional extension in these proteins and an evolutionarily conserved targeting of LDHBx to peroxisomes in mammals (**Figure 9**).

## Piggy-back co-import of LDHA with LDHB

LDHB together with lactate dehydrogenase A (LDHA) can form five tetrameric LDH isoforms, of which two are homotetramers and three are heterotetramers (**Boyer et al., 1963**; **Markert, 1963**), and peroxisomes have the unusual ability to import folded and even oligomeric proteins (**McNew and Goodman, 1996**; **Lanyon-Hogg et al., 2010**). We therefore wanted to test if peroxisomal LDHBx piggy-backs LDHA into peroxisomes. For this purpose we adapted a two-hybrid assay previously used to analyze co-import of subunits of the dimeric peroxisomal hydrolase Lpx1 in a heterologous system (**Thoms et al., 2011**). When LDHA was expressed as a fusion protein with N-terminal YFP without co-expression of any form of LDHB, the protein localized to the cytosol as expected (**Figure 10A**). However, when we co-expressed YFP-LDHA with CFP-LDHBx[TGG], that is cyan fluorescent protein (CFP) fused to the readthrough form of LDHB, we found YFP-LDHA in peroxisomes (**Figure 10B**). This experiment shows that the readthrough form of LDHB, LDHBx, can interact with LDHA, and that LDHBx is capable of carrying LDHA into the peroxisome. To show that co-import of LDHA is dependent on the hidden targeting signal in LDHBx, we mutated the targeting signal to SSI, or we deleted the terminal leucine. Either LDHBx PTS1 mutation blocked co-import of LDHA (**Figure 10—figure supplement 1**). The peroxisome is thus accessible to all four new LDH isoforms containing LDHBx.

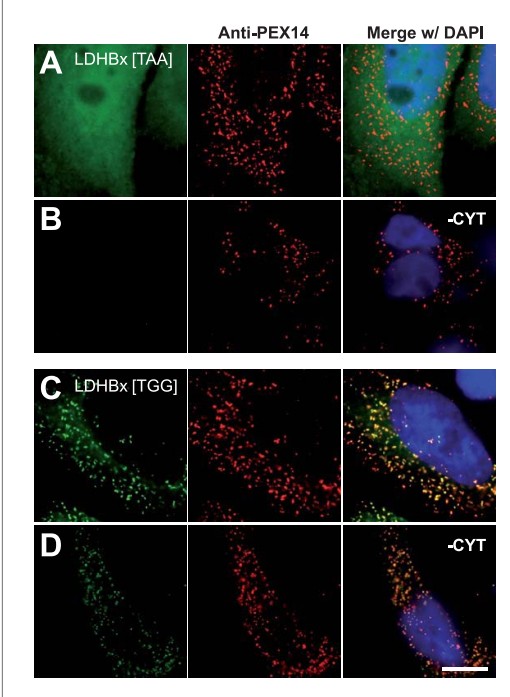

**Figure 6**. Peroxisome targeting of LDHBx is dependent on the stop codon. Combined direct fluorescence and immunofluorescence in HeLa cells. (**A** and **B**) Exchange of UGA stop codon with the tighter stop UAA (YFP-LDHBx[TAA]) reduces peroxisomal localization of LDHB. (**C** and **D**) When UGA is replaced by tryptophan-coding UGG (LDHBx[TGG]), a larger proportion of LDHB is targeted to the peroxisome, and peroxisome localization becomes obvious without removal of the cytosol. (**B**, **D**) Cytosol was removed after cell permeabilization with digitonin. Bar 10 μm.

To support our data on LDHBx-LDHA co-import, we drew a structural model of the LDH-1 tetramer, the fundamental all-B isoform of LDH (*Figure 10—figure supplement 2*). The C-terminal amino acid leucine is extended by three amino acids not resolved in the structure, and, in LDHBx, by an additional seven amino acids. The model shows that this extension protrudes from the tetramer and is located distal to the protomer-interaction site, confirming that oligomerization is not hampered by the extension. The protruding LDHBx extension carrying the PTS1 is also accessible on the tetramer surface for PEX5 binding and import into the peroxisome.

## Discussion

The study of translational readthrough goes back to the origins of molecular biology, but mammalian genes undergoing readthrough have only recently come into focus and are being identified by systemic approaches (*Jungreis et al., 2011*; *Dunn et al., 2013*; *Eswarappa et al., 2014*; *Loughran et al., 2014*). Translational readthrough can be controlled by cis-acting elements, RNA structures of the transcript, that, often mediated by trans-factors, influence the termination process (*Firth et al., 2011*; *Eswarappa et al., 2014*). This mechanism has been termed programmed translational readthrough (PTR) (*Eswarappa et al., 2014*). It is known, however, that the stop codon together with the preceding and immediately following nucleotides (SCC) also influence translational readthrough. We have termed this process basal translational readthrough (BTR) to distinguish it from PTR in general, and also from pharmacologically induced readthrough. In this study we derive a motif conferring high BTR from a linear regression model of SCCs and show that LDHBx undergoes BTR, which in turn affects the intracellular distribution of LDH.

### A new LDH subunit

LDH is an enzyme with several isoforms, which has also been instrumental in devising the enzyme isoform concept per se. The identification of the classic muscle and heart subunits LDH-M (LDHA) and LDH-H (LDHB) in the late 1950s was followed by the identification of a testes-specific LDHA variant, LDHC (*Boyer et al., 1963*; *Goldberg et al., 2009*). Now we find that readthrough-extended LDHBx is encoded by the well-known *LDHB* gene by translational stop suppression and can give rise to new isoforms. Peroxisomal LDH is a novel isoform of LDH containing at least one readthrough-extended LDHBx subunit. LDHB readthrough and readthrough-dependent peroxisomal localization are evident in various human cell types, suggesting that the LDHBx subunit is expressed and localized to peroxisomes in all tissues that express LDHB. LDHBx exemplifies a new mechanism of post-transcriptional diversification of the genome's coding potential in mammals.

The 1.6% LDHBx stop codon readthrough that we find in our experiments corresponds to the 1.5–2% LDH activity found in association with peroxisomes (*McGroarty et al., 1974*; *Osmundsen, 1982*; *Baumgart et al., 1996*), suggesting that cellular suppression of the stop codon is the only pathway for LDHB into peroxisomes. Assuming that peroxisomes fill approximately 1–2% of the cell volume, translational readthrough ensures almost equal concentration of LDH in cytosol and in peroxisomes.

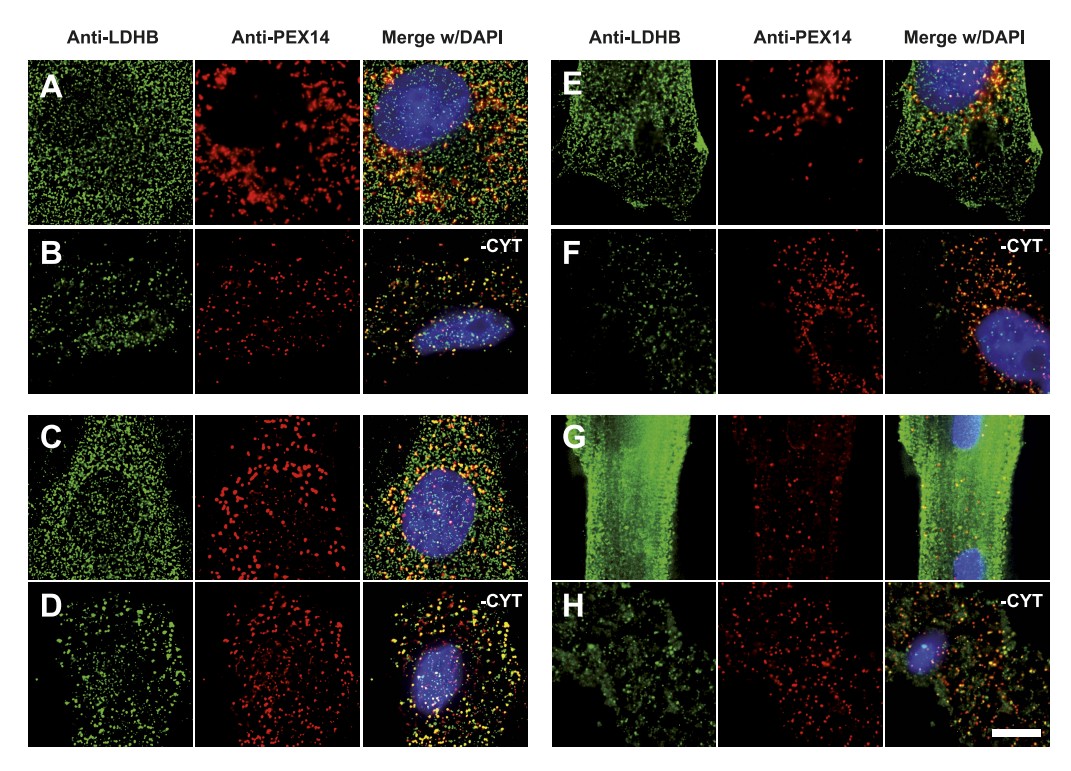

**Figure 7**. Endogenous LDHB is localized to peroxisomes in wild-type cells. Immunofluorescence in wild-type cultured cells (**A**–**F**) or freshly prepared (**G** and **H**) cells with antibodies recognizing LDHB (secondary antibody Alexa488-coupled) and the peroxisome marker PEX14 (secondary antibody Cy3-coupled). (**A** and **B**) COS-7 cells, (**C** and **D**) human skin fibroblasts, (**E** and **F**) U118 glioblastome, and (**G** and **H**) primary rat cardiomyocytes. (**B**, **D**, **F**, **H**) Cytosol was removed after permeabilization with digitonin (-CYT). Bar 10 µm.

## A role for peroxisomal LDH

Fatty acid β-oxidation reactions are the hallmark of peroxisomes in most cell types and organisms. In mammalian peroxisomes, β-oxidation is involved in the degradation of very long chain fatty acids (VLCFA) and biogenetic reactions such as the synthesis of bile acids (*Lodhi and Semenkovich, 2014*). Therefore patients with peroxisomal disorders accumulate VLCFA and bile acid intermediates (*Braverman et al., 2013*). During fatty acid oxidation and other peroxisomal processes, nicotinamide adenine dinucleotide (NAD+) is reduced to NADH. However, the pathway of NAD+ regeneration inside peroxisomes is not clear (*Kunze and Hartig, 2013*). For efficient β-oxidation to occur, it is necessary that a redox shuttle system exists for NAD+ regeneration, because peroxisomes are impermeable to NAD+/NADH (*Visser et al., 2007*). The identification of LDH inside the peroxisome suggested the existence of a lactate/pyruvate shuttle involved in the regeneration of redox equivalents (*Baumgart et al., 1996*; *McClelland et al., 2003*; *Gladden, 2004*). In the absence of a peroxisomal targeting signal, however, peroxisomal LDH was not universally accepted by researchers.

Lactate/pyruvate shuttling could either occur directly through the peroxisomal membrane (*Visser et al., 2007*) or make use of monocarboxylate shuttles in the peroxisomal membrane (*McClelland et al., 2003*). Generally, functional LDHBx targeting to peroxisomes highlights the role of intracellular lactate shuttle mechanisms (*Brooks, 2009*). In liver peroxisomes, pyruvate production is catalyzed by alanine-glyoxylate aminotransferase, an important enzyme in glyoxylate detoxification. Glyoxylate, however, is itself a substrate of LDH (*Salido et al., 2012*). Therefore, peroxisomal LDH may also be involved in peroxisomal glyoxylate metabolism.

Peroxisomal LDH is not the first glycolytic enzyme found in peroxisomes. Trypanosomes have sequestered the full set of glycolytic enzymes in specialized peroxisomes called glycosomes (*Gualdrón-López et al., 2012*). And recently, in fungi, part of the glycolytic pathway upstream of pyruvate including

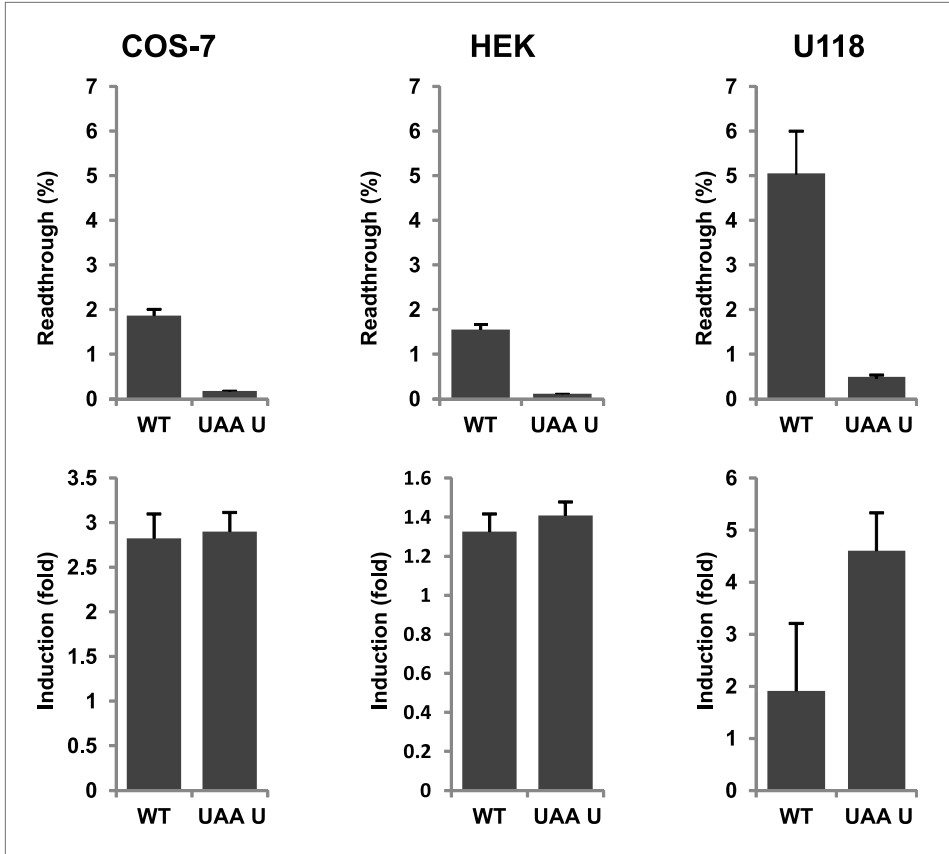

**Figure 8**. Evidence for regulation of readthrough. LDHB stop codon readthrough in various mammalian cell types. COS-7, HEK, and U118 cells were transfected with LDHB and mutant dual reporter constructs and analyzed by Venus fluorescence and luciferase assays. Readthrough is expressed as hRLuc/Venus signal. Readthrough is induced by 100 µg/ml geneticin.

glyceraldehyde-3-phosphate dehydrogenase and 3-phosphoglycerate kinase, was shown to be localized to peroxisomes by alternative splicing and/or translational readthrough (*Freitag et al., 2012*). It is compelling that fungi as well as mammals use stop codon suppression to localize a small fraction of glycolytic enzymes to peroxisomes. We hypothesize that both translational readthrough as well as PTS1 evolve easily, and so can divert a low and steady amount of these enzymes to peroxisomes.

A small fraction of cytosolic LDHB is imported into peroxisomes. This fraction is likely to be constant with respect to the overall LDHB expression levels in given tissue. We speculate that the peroxisomal LDHB shunt helps to coordinate redox processes between the cytosol and the peroxisome. Importantly, our study reveals a new pharmacological effect of readthrough-inducing drugs such as the commonly prescribed aminoglycosides, as they will increase LDHB readthrough and peroxisome import of LDHBx.

It is not known at the moment whether translational readthrough is regulated in humans. The very high readthrough of approximately 5% in a glioblastoma cell line suggests that readthrough is differentially regulated in different tissues. Future experiments will show if the increased LDHB readthrough we find in this cell line are a cancer-associated dysregulation linked to the Warburg effect (*Hsu and Sabatini, 2008*), or if it just matches a higher abundance of peroxisomes in these cells to ensure an equal concentration of LDH in cytosol and peroxisomes in these cells as suggested above. It is also possible that glial cells generally have a higher demand for peroxisomal LDH that could be involved in neuronal/glial lactate metabolism.

## A rational approach to translational readthrough

The first mammalian readthrough proteins were identified by chance (*Geller and Rich, 1980*; *Chittum et al., 1998*; *Yamaguchi et al., 2012*). Recently, two powerful and complementary methods have been

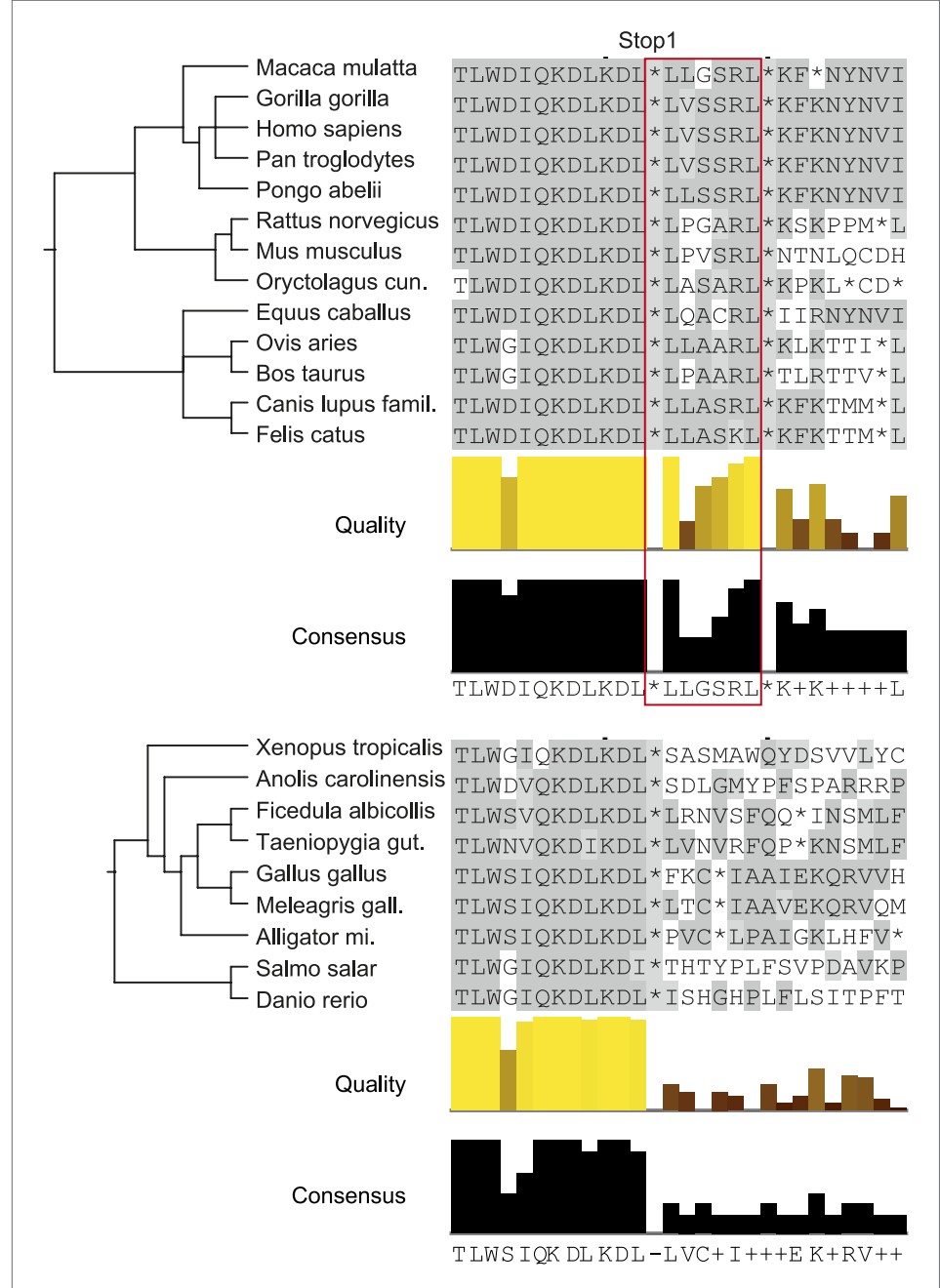

**Figure 9**. LDHBx extensions including hidden PTS1 are strictly conserved in mammals. Alignments of LDHBx termini from mammals and non-mammalian vertebrates. PTS1 extension is boxed. The conserved readthrough PTS1 extension is found exclusively in mammals and marks the mammalian–non-mammalian border in vertebrates. *Alligator mi.*: *Alligator mississippiensis*; *Canis lupus famil.*: *Canis lupus familiaris*; *Meleagris gall.*: *Meleagris gallopavo*; *Oryctolagus cun.*: *Oryctolagus cuniculus*; *Taeniopygia gut.*: *Taeniopygia guttata*.

employed in the genome-wide identification of readthrough-extended proteins. Ribosome profiling can recognize translating ribosomes in 3′UTRs and thereby identify readthrough and other recoding events outside known coding regions (*Ingolia et al., 2011*; *Dunn et al., 2013*). Phylogenetic approaches such as those implemented in PhyloCSF (*Lin et al., 2011*) evaluate the coding potential of sequences before and after the stop codon to help predict readthrough and are particularly powerful when genome sequences from closely related species are available (*Jungreis et al., 2011*; *Loughran et al., 2014*).

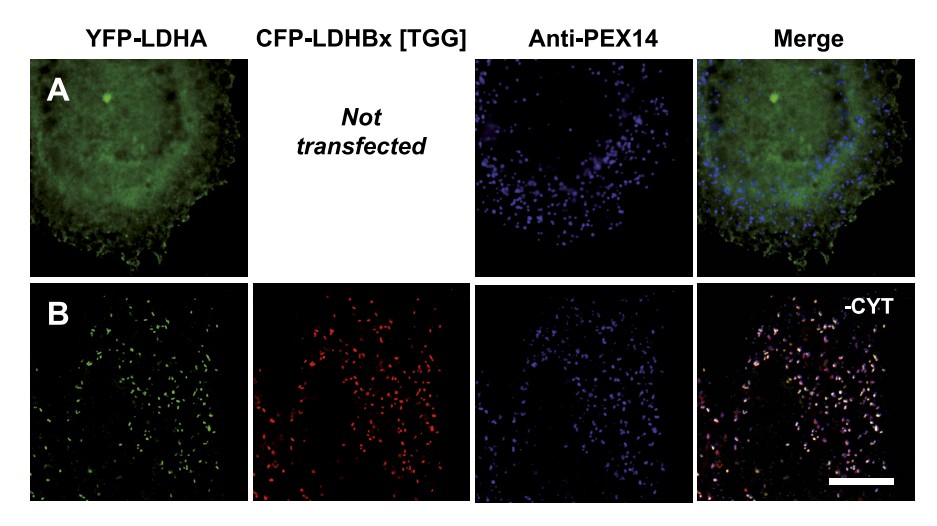

**Figure 10**. Piggy-back co-import of LDHA by LDHBx into peroxisomes. Direct fluorescence of YFP-labeled LDHA (green) in the absence or presence of CFP-labeled LDHBx[TGG] (red) combined with immunofluorescence with a peroxisome marker (blue). (**A**) YFP-LDHA localization is mainly in the cytosol when expressed in the absence of LDHBx. (**B**) LDHA is imported into peroxisomes when co-expressed with LDHBx[TGG]. Cytosol was removed after permeabilization with digitonin. Bar 10 μm.

The following figure supplements are available for figure 10:

**Figure supplement 1**. Mutation of the cryptic targeting signal SRL in LDHBx blocks co-import of LDHA into peroxisomes.

**Figure supplement 2**. Tetrameric lactate dehydrogenase (space fill model) from human heart (LDH-1, all-B isoform).

Ribosome profiling, however, depends on gene expression, and can identify readthrough events only when the cell type in question is actually analyzed. Ribosome profiling may also fail to identify short readthrough extensions. Phylogenetic approaches, on the other hand, may miss readthrough when it is not conserved in the given dataset or when sufficiently dense datasets are not available, or when the extensions are too short to provide a basis for phylogenetic comparison.

Our approach to systems-level identification of translational readthrough is based on the formalization of SCCs and a linear regression model with experimental readthrough values. The majority of the input sequences have been derived from patient nonsense mutations. In consequence, these sequences are biased neither by preselection by any pre-determined RTP or experimental readthrough levels, or by the SCCs, because the contexts did not evolve together with the respective stop codons. The algorithm we develop in this paper is limited to six nucleotide positions before and after the stop codon. This approach excludes the identification of extended RNA secondary structures involved in PTR and other recoding events (*Baranov et al., 2002*; *Firth and Brierley, 2012*; *Eswarappa et al., 2014*; *Loughran et al., 2014*). The identification of the LINfs3 consensus and the human genes associated with this consensus justifies this approach. The LINfs3 motif, derived by feature selection, encompasses the stop codon and the first codon after the stop: UGA CUA. Our analysis suggests that positions +7 and −6 might also further contribute to readthrough. We have tested five of the 144 candidates in the genome with the UGA CUA motif and confirmed their high BTR. Highest BTR appears to correlate with a G in position +7 (UGA CUA G) within the LINfs5 consensus. This motif is found 30 times in the human genome and has recently been shown to support high translational readthrough (*Loughran et al., 2014*). The motifs for high BTR are distinct from the consensus UGA CAR YYA (R = A/G, Y = C/U) found in some viruses and yeast (*Namy et al., 2001*; *Harrell et al., 2002*) but resembles the alphavirus-like high readthrough stop codon context (*Li and Rice, 1993*). Interestingly, the same stop suppression context in the *LAMA3* gene has been shown to alleviate the disease severity of an otherwise fatal nonsense mutation in a patient with junctional epidermolysis bullosa, the major and most devastating form of epidermolysis bullosa (*Pacho et al., 2011*).

The existence of the consensus motif UGA CUA is the origin of the non-linear contribution to RTP in our models. This is supported by the finding that correlation of BTR and RTP for LINfs3 is higher than for the LINiter model, so that the reduced number of parameters in LINfs3 provides a better model fit. This finding implies that with the currently small dataset, compact linear models should be preferred over non-linear models with many parameters. The identification of the few relevant nucleotide positions will help to create datasets with fully specified BTR for a wide range of SCCs and cell types. A larger training set of sequences with verified readthrough rates will allow the development of non-linear approximation models.

LDHBx shows an unusually high readthrough of 1.6%, and its stop context UGA CUA G (stop codon underlined) matches the LINfs3 consensus. The 18-nucleotide extension in LDHBx is unlikely to contain a extensive secondary structure that would suggest a combined effect of BTR and PTR. The identification of LDHBx and the recently discovered readthrough form of vascular endothelial growth factor A, VEGF-Ax (*Eswarappa et al., 2014*), thus mark two extreme and separable cases of physiological stop suppression: LDHBx appears independent of cis-factors beyond the SCC and marks a prototypical example of BTR. In contrast, the readthrough of VEGF-Ax is relatively independent of its SCC but instead requires a more distantly located cis-element (*Eswarappa et al., 2014*). The distinction between PTR and BTR, however, is not exclusive. A thorough analysis of readthrough in OPRK1 and OPRL1 indicates that readthrough levels of more than 30% can be obtained by a combination of cis-elements and UGA CUA-based BTR (*Loughran et al., 2014*).

The era of systematic analysis of translational readthrough in humans is only beginning. We expect that a combination of in silico modeling and screening, ribosome profiling, phylogenetic methods, and mass spectrometry will help to identify the 'extensome', the complete set of readthrough-extended proteins in mammals.

## Materials and methods

### RTP calculation algorithm

To predict the RTP of gene transcripts, we developed a linear regression model based on the SCCs and their experimentally determined basal readthrough values. The SCC comprises the stop codon itself (positions +1 to +3) and the nucleotide sequences surrounding the stop codon (−6 to +9). For the first-pass model (LIN), we re-analyzed 66 SCCs with known experimental basal readthrough values (*Floquet et al., 2012*). The stop codons evolved independently of their contexts (*Table 3*). Nucleotide sequences were represented by indicator vector coding. Here, 12 × 4 binary vector entries are used to indicate the presence [1] or absence [0] of a nucleotide (A, C, G, or U) at a particular position (−6 to −1, +4 to +9) surrounding the stop codon. Three further entries are reserved to indicate the type of stop codon (UAA, UAG, or UGA, positions +1, +2, or +3). The resulting feature vectors of all sequences were normalized to Euclidean unit length.

For the estimation of the regression model coefficients, we performed a regularized least-squares ('rigde') regression (*Hoerl and Kennard, 1970*). Let $X$ be the $n \times d$ matrix of $n$ sequence feature vectors with dimensionality $d$ and $y$ be the ($n$-dimensional) vector of readthrough values associated with the sequences. Then the weight vector $w = (X^T X + k \times I)^{-1} \times X^T y$ represents the solution of the linear least-squares problem and $y = w^T x$ corresponds to the RTP value $y$ for a sequence feature vector $x$. To evaluate the influence of the regularization parameter $k$, we performed a leave-one-out cross-validation (loo-cv) with $k = \{10^i | i = -3, -2.7, ...0,...,3\}$ for all model types. The minimum loo-cv error in terms of the sum of squared deviations of predictions from known readthrough values was $4.75 \times 10^{-7}$ for $k = 10^{0.3}$ (approximately 1.995).

For genome-wide prediction of readthrough propensities for human transcripts, we downloaded all 215,621 coding sequences from the Ensembl BioMart (*Flicek et al., 2012*) using the Homo sapiens Genes v74 section (GRCh37.p13) plus 300 nucleotides downstream of the CDS end (ensembl.org, November 2013). Transcripts corresponding to identical protein products, short sequences (<15aa protein-coding) and incomplete (e.g., missing or mislocated stop codon) or insufficiently sequenced (i.e., undetermined nucleotides) DNA were removed. Sequences with identical 3'/C-termini (nucleotide positions −45 to +303) were aggregated to one representative sequence, resulting in 42,069 unique transcripts. ORF extensions were identified by detection of an in-frame stop codon within 300 nucleotides downstream of the annotated stop codon.

**Table 3.** Nucleotide frequencies in each position of the stop codon context

| Nucleotide | A | C | G | U |
|---|---|---|---|---|
| Position | | | | |
| −6 | 0.2892 | 0.2530 | 0.2651 | 0.1928 |
| −5 | 0.3253 | 0.2651 | 0.1446 | 0.2651 |
| −4 | 0.1566 | 0.2289 | 0.3494 | 0.2651 |
| −3 | 0.2410 | 0.3373 | 0.2410 | 0.1807 |
| −2 | 0.2651 | 0.1807 | 0.2048 | 0.3494 |
| −1 | 0.2410 | 0.2530 | 0.2651 | 0.2410 |
| 4 | 0.2289 | 0.3133 | 0.3373 | 0.1205 |
| 5 | 0.2651 | 0.2530 | 0.1446 | 0.3373 |
| 6 | 0.2771 | 0.2169 | 0.2530 | 0.2530 |
| 7 | 0.2530 | 0.3133 | 0.2892 | 0.1446 |
| 8 | 0.3253 | 0.1687 | 0.2410 | 0.2651 |
| 9 | 0.1807 | 0.2771 | 0.2771 | 0.2651 |
| Stop codons | UAA | UAG | UGA | |
| 1 to 3 | 0.1928 | 0.3373 | 0.4699 | |

The nucleotide and stop codon frequencies for positions −6 to −1 and 4 to 9 were calculated for the 81 sequences used in the RTP predictor (LINiter model).

## Iterative model refinement and feature selection

To obtain a more comprehensive model for RTP prediction, we included 15 sequences and their corresponding experimentally determined readthrough values from this study in the prediction model (see *Schueren et al., 2014* for Dataset 1). The regression coefficients for the iterative model considering all 12 stop context positions (LINiter) were computed as described in the previous section. The minimum regression error was $6.24 \times 10^{-6}$ at $k = 10^{0.3}$. A sequence logo representation of the regression coefficients for this model is displayed in *Figure 2A*. The sequence logo was created using the enoLOGOS web server (*Workman et al., 2005*).

Furthermore, we evaluated reduced model sizes by stepwise elimination of context positions carrying no or little information for RTP prediction (feature selection). Starting from the complete mode (LIN), we removed the position corresponding to the minimum sum of squared regression coefficients. Regression error and coefficients were then calculated for the remaining positions (including the stop codon) as described above. This procedure was repeated until only the stop codon position was left. *Figure 2B* shows the development of the regression error for reduced model sizes by stepwise elimination of positions. Here, a first local minimum can be identified for model LINfs5 with five positions remaining (−6, stop, +4 to +7) and the global minimum corresponds to model LINfs3 with three positions besides the stop codon (stop, +4 to +6).

## PTS1 prediction algorithm

To identify cryptic peroxisomal localization signals in readthrough extensions, we adapted a peroxisomal targeting signal type 1 (PTS1) detection algorithm that was previously developed for plant proteins (*Lingner et al., 2011*). For this purpose, we used 24 known human PTS1 proteins (ACOT4, ACOX1, ACOX2, ACOX3, AGXT, AMACR, BAAT, CRAT, DAO, EHHADH, GNPAT, HAO1, HAO2, HSD17B4, IDE, MLYCD, PRDX5, ACOT8, CROT, PECI, ECH1, LONP2, PECR, and PIPOX) and performed orthology searches on metazoan protein and EST sequences using a bidirectional best BLAST hit strategy. Starting from each human protein sequence, we identified significant BLAST hits (e-value < $10^{-10}$) to metazoan sequences within the 'nr' and 'dbEST' database. Then, the best hit of each organism was searched against the human proteome and sequences not re-identifying the starting sequence were removed. Afterwards, the starting sequences and putative orthologs were pooled and sequences with uncommon PTS1 tripeptides, that is tripeptides which occurred less than three times, were removed from the set. The resulting set of sequences was used as positive examples

for training machine learning models as previously published (*Lingner et al., 2011*). Briefly, a regularized least-square classification algorithm was trained using indicator vector representations of up to 15 C-terminal amino acids of positive and negative example sequences. A set of negative example sequences was created by extracting all metazoan sequences without peroxisomal association from the Swiss-Prot section of UniProt (http://www.uniprot.org) in November 2011. The best model (15 C-terminal amino acids) was determined by fivefold cross-validation and yielded a prediction accuracy of 0.996 and 0.863 in terms of the area under the receiver operating characteristic (ROC) curve (auROC) and the area under the precision/recall curve (auPRC), respectively. When a stop codon was considered in the PTS1 prediction, the stop codon was scored as an undefined amino acid ('X') without a contribution to the PTS1 posterior probability.

## Multiple alignment analysis

The multiple alignment of genomic sequences for the LDHB SCC (position −36 to +48) was downloaded from the Ensembl database (www.ensembl.org) in November 2013. The '21 amniota vertebrates' alignment was used and split into mammalian and non-mammalian species. Sequences without residues in the extension region were deleted and the non-mammalian alignment was augmented by LDHB sequences from the NCBI nucleotide database (http://www.ncbi.nlm.nih.gov/nuccore) in November 2013. In total, the alignments comprise 13 mammals and nine non-mammalian vertebrates: *Homo sapiens* (human), *Mus musculus* (mouse), *Rattus norvegicus* (rat), *Oryctolagus cuniculus* (rabbit), *Pan troglodytes* (chimpanzee), *Gorilla gorilla* (gorilla), *Pongo abelii* (orangutan), *Macaca mulatta* (rhesus macaque), *Felis catus* (cat), *Canis familiaris* (dog), *Equus caballus* (horse), *Bos taurus* (cow), *Ovis aries* (sheep), *Xenopus tropicalis* (western clawed frog), *Anolis carolinensis* (anole lizard), *Ficedula albicollis* (flycatcher), *Taeniopygia guttata* (zebra finch), *Gallus gallus* (chicken), *Meleagris gallopavo* (turkey), *Alligator mississippiensis* (American alligator), *Salmo salar* (salmon), and *Danio rerio* (zebrafish).

The genomic sequences were translated into amino acid sequences using the 'EMBOSS Transeq' web server (http://www.ebi.ac.uk/Tools/st/emboss_transeq/). Species trees were obtained from the Interactive Tree Of Life (iTOL) website (http://itol.embl.de/) and visualized with the Phylip package (*Felsenstein, 1989*). JalView software (*Waterhouse et al., 2009*) was used to visualize the alignments and to compute alignment quality and consensus. Here, the quality score of an alignment column is inversely proportional to the average cost of all pairs of mutations in terms of BLOSUM 62 substitution scores and the consensus reflects the fraction of the most frequent residue for each column of the alignment.

## DNA cloning

Plasmids used in this study are listed in the table in *Supplementary file 1*. Oligonucleotides used in this study are listed in the table in *Supplementary file 2*.

The dual reporter vector pDRVL (PST1360) encoding an N-terminal Venus tag and a C-terminal hRluc tag was derived from pEXP-Venus-hRluc (a gift from Ania Muntau and Sören Gersting) by introducing a short multicloning site (MCS) containing BstEII, ClaI, BspEI, and BsiWI restriction sites. pDRVL was created by ligating pre-annealed oligonucleotides OST963 and OST964 into the XhoI site of pEXP Venus-hRluc. Dual reporter constructs PST1384–1385, 1387, 1393–1396, 1418–1426, 1430, 1435, 1437, 1493, 1494, 1497, 1504, and PST1444 were derived from pDRVL by insertion of pre-annealed oligonucleotides OST1081–1084, 1086–1087, 1117–1124, 1144–1145, 1148–1157, 1160–1165, 1158–1159, 1190–1191, 1198–1199, 1229–1230, JH59–60, JH61–62, JH67–68, and JH81–82 into BspEI and BstEII sites, as listed in *Supplementary file 2*.

For cloning of pEYFP-LDHBx (PST1388), the LDHB open reading frame including the stop codon and the 18-nucleotide 3' extension, was PCR-amplified from pOTB7-LDHB using primers OST1053 and 1054 and inserted into EcoRI and XbaI sites of pEYFP-C1.

The stop codon variants pEYFP-LDHBx[TGG] (PST1389), pECFP-LDHBx[TGG] (PST1440), pEYFP-LDHBx[TAA] (PST1410), pEYFP-LDHBx[TAAT] (PST1411), and pEYFP-LDHBx[TGAT] (PST1409) were created by amplifying LDHBx using primer OST1053 with reverse primers OST1055, 1127, 1128, and 1129, respectively. Similarly, the PTS1 mutation variants pEYFP-LDHBx[ΔL] (PST1407), pECFP-LDHBx[TGG, ΔL] (PST1512) (deletion of the last amino acid in the cryptic PTS1 SRL), and pEYFP-LDHBx[SSI] (PST1408), pECFP-LDHBx[TGG, SSI] (PST1513) (substitution of the PTS1 SRL by SSI) were created using forward primer OST1053 and reverse primers OST1125, 1263, 1126, and 1264, respectively. LDHA was amplified from human cDNA using primers OST1130 and 1131 and cloned into EcoRI and XbaI sites of pEYFP-C1 to yield pEYFP-LDHA (PST1434).

For cloning of pEXP Venus-PTS1 (PST1209), primers OST801 and 802 (encoding the PTS1 of ACOX3) were annealed and inserted into pENTR-TOPO-D. Then the PTS1 tag was transferred to pEXP-N-Venus using LR clonase II (Invitrogen, Carlsbad, California).

Full-length dual reporter constructs pcDNA3.1-HA-LDHBx-Myc and variants were cloned by amplifying LDHB and stop codon variants from PST1388 (LDHB wt), PST1389 (LDHB [TGG]), PST1409 (LDHB [TGAT]), PST1410 (LDHB [TAA]), and PST1411 (LDHB [TAAT]), using primers OST1202 and 1203 and cloning into NheI and BamHI restriction sites of pcDNA3.1/Myc-His(−)A. All plasmids were confirmed by DNA sequencing.

## Cell culture and transfection

HeLa cells and human skin fibroblasts were maintained in low glucose Dulbecco's minimal essential medium (DMEM), HEK cells, HT1080, U118, U373 and COS-7 cells in high glucose DMEM. Culture media were supplemented with 1% (wt/vol) glutamine, 5–10% (vol/vol) heat inactivated fetal calf serum (FCS), 100 units/ml penicillin, and 100 µg/ml streptomycin. For U118 cells, 1% non-essential amino acids and 1% pyruvate were added to the media.

Cells were transfected using Effectene transfection reagent (Qiagen, Germany) as described by the manufacturer. Plasmids were diluted in Buffer EC and Enhancer and incubated for 5 min at room temperature. Effectene was added and incubated for 10 min at room temperature. Prewarmed medium was added to the HeLa cells and to the transfection mixture which was then added to cells and incubated at 37°C in a humidified 5% $CO_2$ incubator for 24 hr. Then, 6 hr after transfection, transfection reagent was removed, and, where indicated, geneticin (G418) was added at a concentration of 100 µg/ml.

## Dual reporter assays and readthrough calculation

Cells were washed with PBS and lysed by Renilla Luciferase Assay Lysis Buffer (Promega, Madison, Wisconsin) according to the manufacturer's manual. Cells were spun down (14 krpm, 2 min, 4°C) and supernatants were stored at −80°C. For Venus fluorescence measurement, cell lysates were diluted 1:25 in PBS and analyzed at 485 nm excitation, 530 nm emission (sensitivity: 130) using a Synergy Mx plate reader (Biotek, Winooski, Vermont). PBS was used as a blank control for fluorescence measurements.

Undiluted lysates (20 µl) were used to measure hRluc luminescence by the Renilla Luciferase Assay System (Promega) and the Synergy Mx plate reader (Biotek). An automated injector was used to add 100 µl Renilla Luciferase Assay Reagent. Luminescence was read 2 s after injection and integrated over 10 s (sensitivity: 150). Renilla Luciferase Assay Reagent was used as a blank control for hRluc luminescence measurements. Each construct was analyzed in three to seven biological replicates and each biological sample was measured in triplets.

To obtain readthrough rates, the ratio of hRluc/Venus fluorescence was calculated, and the readthrough of pDRVL was set to 100%. The ratio (y) and standard deviation of fluorescence ($x_1$) and luminescence ($x_2$) signal for each replicate were calculated using uncertainty propagation ($\sigma_y = [\sigma^2_{x1} \times (dy/dx_1)^2 + \sigma^2_{x2} \times (dy/dx_2)^2]^{0.5}$). Let $w_i = 1/\sigma_i^2$ be the weight of a readthrough value from replicate i with $\sigma_i$ being the error of the ratios. Then the weighted mean $x_m$ of the replicates and its error $\sigma_{xm}$ were calculated according to $x_m = (\Sigma_i(x_iw_i)/\Sigma_iw_i)$ and $\sigma_{xm} = (\Sigma_iw_i)^{-0.5}$.

## Immunofluorescence, microscopy, and quantification

Transfected LDHB and LDHA fusion constructs were detected in HeLa cells by combined direct fluorescence and immunofluorescence experiments. Endogenous LDHB was analyzed in HeLa, U118, and COS-7 cells, and in primary rat cardiomyocytes by immunofluorescence. Approximately $1 \times 10^5$ cells were seeded on cover slips or on laminin-coated (Sigma, St. Louis, Missouri) glass slides for HEK cells and cardiomyocytes and transfected as indicated. For removal of cytosol, cells were treated with 0.02% (wt/vol) digitonin (Invitrogen) for 5 min at room temperature. Cells were fixed with 10% (wt/vol) formaldehyde for 20 min, and permeabilized with 0.5% Triton X-100 for 5 min. After blocking for 20 min at 37°C with 10% BSA, antigens were labeled with primary antibodies at 37°C for 1 hr. Antibody dilutions were 1:200 for anti-PEX14 rabbit polyclonal antibodies (ProteinTech, Chicago, Illinois) and 1:500 for anti-LDHB mouse monoclonal antibodies (Abnova, Taiwan). Secondary antibody labeling (1:200) was done for 1 hr with antibodies labeled with Cy3 and/or Alexa647 (Jackson Immuno Research, West Grove, Pennsylvania) and/or Alexa488 (MoBiTech, Germany). Cover slips were mounted with Mowiol containing 0.01 mg/ml 4′,6-diamidino-2-phenylindole (DAPI). DAPI was omitted in cases where cells had been transfected with CFP-expressing plasmids.

Fluorescence microscopy was done using a 100× oil objective (1.3 NA) with a Zeiss Imager M1 fluorescence wide field scope equipped with the Zeiss Axiocam HRm Camera and Zeiss Axiovision 4.8 acquisition software. z-Stacks with 30 images and 0.25 µm spacing were recorded and subjected to deconvolution. Where necessary, linear contrast enhancements were applied (Axiovision).

To quantify induction of endogenous LDHB by geneticin, fluorescence images from samples prepared with anti-LDHB and anti-PEX14 antibodies were recorded under identical conditions and subjected to deconvolution. The LDHB/PEX14 intensities were measured, and the same threshold ratios were applied to all channel pairs (ImageJ). Induction is expressed as the ratio of LDHB/PEX14 ratios with and without geneticin treatment, respectively.

### Western blot analysis

Cells were lysed in RIPA lysis buffer (20 mM Tris–HCl, pH 7.4, 150 mM sodium chloride, 2 mM EDTA, 1% NP40, 1 mM DTT, 0.1 mM PMSF, Complete protease inhibitors [Roche, Switzerland]) 24 hr after transfection. Proteins were separated by SDS-PAGE on a 12% gel, transferred to a nitrocellulose membrane, and probed with primary and secondary antibodies. The following antibodies were used: anti-HA rabbit polyclonal (Abcam, UK), anti-Myc mouse monoclonal (Cell Signaling, UK), anti-luciferase mouse monoclonal (Millipore), anti-GFP mouse monoclonal (Living Colors, Mountain View, California), and anti-actin mouse monoclonal (Sigma). HRP-conjugated goat anti-rabbit IgG and donkey anti-mouse IgG (Jackson Immuno Research) were used as secondary antibodies. We also used 1:1000 dilutions of primary antibody and 1:5000 dilutions of secondary antibody. Reactive bands were revealed with Lumi-light and Lumi-light plus Western blotting substrate (Roche). Images were scanned using Luminescent image analyzer LAS 4000.

### Data availability

Dataset 1. Spreadsheet containing predicted readthrough extensions, RTP scores (LIN, LINiter, LINfs5, LINfs3), PTS1 scores, predictions of ER retentions signals, glycosylation motifs, transmembrane domains, and transmembrane topology, and the LINiter$^+$ × PTS1 product scores for all human transcript termini. Publicly available at the Dryad Digital Repository with the doi 10.5061/dryad.j2n18 (*Schueren et al., 2014*).

## Acknowledgements

We thank Heiner Klingenberg for help with orthology searches of human PTS1 proteins, Ania Muntau and Sören Gersting for plasmids, and Kristina Gamper and Viacheslav Nikolaev for the rat cardiomyocytes. We are grateful to Ellen Krämer and Tanja Wilke for technical assistance, and to Cindy Krause, Peter Meinicke, Olaf Jahn, Johannes Freitag, and Michael Bölker for discussions. We thank Blanche Schwappach, Heinz Neumann, Heike Krebber, and Maya Schuldiner for comments on the manuscript.

## Additional information

### Funding

| Funder | Grant reference number | Author |
| --- | --- | --- |
| Georg-August-Universität Göttingen | Research Program, Faculty of Medicine, | Sven Thoms |
| Deutsche Forschungsgemeinschaft | LI2050/1-1 | Thomas Lingner |
| Deutsche Forschungsgemeinschaft | GA354/7-1 | Jutta Gärtner |

The funders had no role in study design, data collection and interpretation, or the decision to submit the work for publication.

### Author contributions

FS, RG, JH, CD, Acquisition of data, Analysis and interpretation of data; TL, Conception and design, Acquisition of data, Analysis and interpretation of data; JG, Drafting or revising the article; ST, Conception and design, Acquisition of data, Analysis and interpretation of data, Drafting or revising the article

## Additional files

### Supplementary files

• Supplementary file 1. Plasmids used in this study.

• Supplementary file 2. Oligonucleotides used in this study.

### Major dataset

The following dataset was generated

| Author(s) | Year | Dataset title | Dataset ID and/or URL | Database, license, and accessibility information |
|---|---|---|---|---|
| Schueren F, Lingner T, George R, Hofhuis J, Dickel C, Gärtner J, Thoms S | 2014 | Data from: Peroxisomal lactate dehydrogenase is generated by translational readthrough in mammals | http://dx.doi.org/10.5061/dryad.j2n18 | Available at Dryad Digital Repository under a CC0 Public Domain Dedication. |

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
