## [Decision Letter]

Thank you for choosing to send your work entitled “Functional translational readthrough in humans” for consideration at *eLife.* Your full submission has been evaluated by James Manley (Senior editor), a Reviewing editor, and 3 peer reviewers, and the decision was reached after discussions between the reviewers. We regret to inform you that the extra work we think will be needed falls outside the scope of a resubmission, but if you can address the concerns below we would be happy to review a new submission at a later date.

You describe the derivation of a new algorithm to deduce the propensity for nonsense codon readthrough. The resulting quantitative predictor, called the RTP score, was initially derived from published data on aminoglycoside-promoted readthrough efficiency and further developed by the use of dual reporter readthrough assays. You find provocative correlations of expression and readthrough, i.e., genes that are highly expressed have a lower RTP score, whereas genes that are expressed at low levels have higher RTP scores. By coupling the RTP predictor with a peroxisomal targeting sequence (PTS) prediction algorithm you sought genes that have high scores for RTP as well as PTS. This approach identified a new isoform of LDHB that arises from the readthrough of the normal termination codon resulting in translation of a C-terminally extended protein product harboring the peroxisomal targeting sequence. Monitoring the subcellular localization of the appropriate fluorescent fusion proteins validated this conclusion.

Overall, this is an interesting manuscript, the experiments are well executed, and many of the conclusions are well supported by the data. However, there are major concerns with the paper. One problem is the fact that the function for LDHBP+ in the peroxisome is only postulated, not formally demonstrated (for example, by a loss-of-function experiment or by clinical data). Another major problem is the molecular mechanism involved (please see below). Another concern relates to the repeated claims that the RTP models are “a quantitative predictor for readthrough propensity in the human genome,” which are not supported by a clear, quantitative analysis of RTP performance. Because RTP is a major focus of the manuscript, you should provide additional analyses to substantiate these claims. These are critical questions you must address in any future new submission to *eLife.*

1) The overall rationale of the authors' algorithms needs to be explained in more elaborate, general terms. Further, there needs to be some indication of “public” availability of the algorithm.

2) Dual reporter assays play a key role in the assessment of nonsense codon contexts that are prone to readthrough. However, it appears that all of these assays used by the authors have the traditional format that excludes any introns. This may be a significant problem since deposition of EJC proteins appears to play a key role in translational efficiency (Wiegand et al [2003] Proc Natl Acad Sci USA 100: 11327-11332; Gudikote et al. [2005] Nature Struct Mol Biol 12: 801-819; Nott et al. [2004] Genes Dev 18: 210-222). The latter experiments raise the possibility that the lack of introns/EJC proteins may affect readthrough efficacy. Hence, at a minimum, the authors need to acknowledge this shortcoming of their experimental approach.

3) The authors make a strong point of being the “first” to observe that nonsense suppression allows two mammalian proteins to be derived from the same mRNA. However, this point is also apparent from the supplementary data presented in Welch et al. (2007) (Nature 447: 87-91). Again, this should be acknowledged.

4) The authors do not discuss/compare the consensus sequence identified from their analyses with any of the stop codon sequence contexts that have been reported in the literature. A comparison with known sequence contexts would be helpful for the readers and may exemplify the strength of the RTP predictor.

5) The authors use one specific case study, the peroxisomal targeting sequence, to show physiological functions of a readthrough protein. However, adding one more example, other than the PTS, would greatly enhance the applicability of the RTP predictor combined with other in silico analyses. Experimental validation of another case study might be beyond the scope of the paper, but it would show a wider applicability of the method described.

6) In the experimental validation of the role of readthrough in the synthesis of LDHB+P, the authors should demonstrate the validity of their RTP predictor by doing mutational analyses of the stop codon context according to their consensus models.

7) Although the authors have an intriguing model for the co-import of LDHA with LDHB, It would be important to show two-hybrid data to support it, or address it biochemically using co-IP analyses of the LDHB+P and LDHA.

8) Although they clearly demonstrate that the extended protein is addressed to the peroxysome it is still not clear what is the function of this protein in this organelle. There is no clear demonstration of a physiological function for this protein. Is there any genetic disease linked either to the absence of the PTS1 motif in LDHB or to the modification of the readthrough consensus motif identified by the authors?

9) Concerning the molecular mechanism involved. The fact that this readthrough is UGA specific is very intriguing. Indeed up to now the context is known to modify readthrough efficiency but not in such extend. We would expect variations but not a total absence of readthrough product with the two other stop codons. This is reminiscent to the insertion of Sel that is also specific of the UGA codon. It seems that this possibility can be excluded due to the absence of a SECIS element, but can you clearly exclude an alternative mechanism such an editing or an alternative splicing? Supplementary data answering these questions will be important to provide.

10) It is unclear whether RTP identifies true readthrough candidates, because there are no obvious negative controls to determine whether the readthrough level is “programmed”.

The authors select fifteen stop codon contexts scored by the lin model, and measure their readthrough rates using an eYFP-stop-luciferase reporter. The observed readthrough rates – with the exceptions of MDH1 (2.91% readthrough) and LDHB (1.55%) are quite low, ranging from 0.1-0.66%. This is a much lower amount of readthrough than is observed for bovine VEGF-Ax (10%, [11]) or various human genes (ranging from 0.7 to 60%, [10]). It is unclear whether this amount of readthrough represents a basal level of readthrough that could be obtained with more or less any stop codon context, or an elevated level of readthrough due to something special about the nucleotide contexts selected by the lin model.

To address this, the authors should include in this assay a set of negative controls, including a construct lacking luciferase, and a set of contexts with very low RTP scores, to establish a background readthrough rate. The authors should then discuss how far above background, if at all, their readthrough candidates are.

11) The authors claim that RTP is a genome-wide predictor of readthrough, but they do not substantiate this claim with genome-wide data. The authors should provide a genome-wide examination of readthrough using some sort of experimental measure. For example, they could estimate readthrough rates genome-wide using published ribosome profiling data, and compare these to the predictions made by RTP.

12) The authors claim that RTP is quantitative, but provide no analysis of its quantitative performance. The authors should provide an explicit analysis demonstrating that RTP is quantitative, or a discussion of why it might not be, especially in light of the fact that they acknowledge that “...there is non-linear contribution to RTP in the sense that other high-RTP genes are likely to show readthrough levels higher than predicted.” Such an analysis might be a quantitative comparison (e.g., a scatter plot and a Pearson correlation coefficient) of predicted and observed readthrough rates for all stop codon contexts tested, or an ROC curve describing the sensitivity and specificity of RTP at different score thresholds. Finally, because RTP is a new metric, descriptive statistics on the RTP score distribution should be provided to facilitate interpretation of individual scores.

13) As mentioned above, the readthrough rates observed for the vast majority of candidates identified in this work are very low compared to known examples. It therefore seems likely that stop codons undergoing higher levels of readthrough were either missed by RTP, or not tested by the experimenters. In addition, certain cases of readthrough are regulated in rodents (59) and flies ([10]; Robinson & Cooley, 1997) or differ between different wild-type yeast strains (Torabi & Kruglyak, 2011). Consistent with these observations, the authors themselves note that LDHB undergoes far more readthrough in human U118 cells (∼5%) compared to human fibroblasts and HEK cells (1-2%). It is therefore possible that these models could be overfit to the cell type in which the experiment was performed, in addition to the specific contexts on which the models were trained. In light of these facts, the authors must establish the scope of their claims.

---

## [Author Response]

We are now submitting a completely revised manuscript. In this paper, we have addressed all of the concerns raised by the reviewers. The changes we made cover both the experimental work and the written material in all sections of the manuscript. Nearly all figures were revised and new figures and figure panels were included (Figure 2, Figure 1—figure supplement 2, Figure 2—figure supplement 1 and Figure 2—figure supplement 2, and Figure 10—figure supplement 1). We have also restructured the distribution of data between the main and supplementary figures in Figures 5 and 10 to achieve a more logical succession of arguments. For the same reason, one figure was split into two (now Figures 5 and 6), and, following one reviewer’s suggestion, we have deleted one figure (formerly Figure 3). Lastly, we have amended the supplementary database by more *in silico* analyses and an “applet” (see below). Julia Hofhuis has been involved in some of the additional experiments and is now added as a co-author.

While our manuscript was under review, Eswarappa et al. [Cell 157, 1405-18 (2014)] reported programmed translational readthrough (PTR) in the vascular endothelial growth factor A. This study gave us the opportunity to explain our intentions and the focus of our work in more detail. In their study, Eswarappa et al. investigate a case of programmed translational readthrough, which is dependent on a 63 nucleotide cis-element in the transcript. Remarkably, this case of PTR is relatively independent on the stop codon and its context. In contrast, our work focuses on the ability of the stop codon and its context (stop codon context, SCC) to stimulate translational readthrough. To distinguish this type of readthrough from PTR, we now define this SCC-dependent readthrough as basal translational readthrough (BTR). BTR differs also from pharmacologically induced readthrough. The distinction between PTR and BTR might help to explain why the experimental readthrough values vary over a large range with PTR reaching levels of 10, 30, or even 60% for some genes, while BTR does not seem to exceed 1.5 to 5%. In the latter case, readthrough is entirely dependent on a natural stop codon and the nucleotides in its immediate vicinity, whereas PTR involves cis-elements and trans-factors that can enhance readthrough to a large extend. 1.5% BTR may appear low from the perspective of a PTR-regulated gene, but considering that stop codons usually permit less than 0.13% readthrough, this is an at least 10-fold increase. And for the case of LDHB+P (which we now term LDHBx) we provide evidence that it has a biological function. The distinction between PTR and BTR is not to imply that both are mutually exclusive. Another recent study by Loughran et al. [NAR; doi 10.1093/nar/gku608 (2014)] investigates cases of what is probably a combination of both. To allude to what is the main focus of the paper, we have changed the title to “Peroxisomal lactate dehydrogenase is generated by translational readthrough in mammals”.

*1) The overall rationale of the authors' algorithms needs to be explained in more elaborate, general terms. Further, there needs to be some indication of “public” availability of the algorithm*.

In the revised version of the manuscript we elaborate in detail the rationale of our *in silico*-approach (see Introduction, Results, and Discussion sections). We also extended the explanations in the Figure legends. We have taken special care to point out that our approach focuses on the nucleotides of the stop codon context (SCC) and can therefore only assess readthrough that is dependent on this relatively short stretch close to the stop codon. We have included the RTP values for all human stop codons (Dataset 1). To further increase the accessibility, we have added the regression coefficients (Table 2), and we add an ‘applet’ to the dataset (Excel sheet containing a set of cell functions) that allows RTP-calculation of user-entered SCCs.

*2) Dual reporter assays play a key role in the assessment of nonsense codon contexts that are prone to readthrough. However, it appears that all of these assays used by the authors have the traditional format that excludes any introns. This may be a significant problem since deposition of EJC proteins appears to play a key role in translational efficiency (Wiegand et al [2003] Proc Natl Acad Sci USA 100: 11327-11332; Gudikote et al. [2005] Nature Struct Mol Biol 12: 801-819; Nott et al. [2004] Genes Dev 18: 210-222). The latter experiments raise the possibility that the lack of introns/EJC proteins may affect readthrough efficacy. Hence, at a minimum, the authors need to acknowledge this shortcoming of their experimental approach*.

We would like to thank the reviewers for pointing this out, and helping us clarify the scope of our study. Indeed, we analyzed SCCs, and all of the RTP calculations are entirely based on the regression coefficients of the few nucleotides of the SCC. We do not take into account that cis-sequences or secondary structures within the extension could contribute to readthrough. As stated above, Eswarappa et al. [Cell 157, 1405-1418 (2014)] recently published complementary work by analyzing one example of programmed translational readthrough (PTR), dependent on a 63 nucleotide cis-element 3’ of the stop codon. In contrast, our work concentrates on basal translational readthrough (BTR), which is defined here as being independent of genetic elements outside of the nucleotides surrounding the stop codon. We have identified an SCC motif that leads to high basal translational readthrough (BTR). We also provide evidence that this element acts independently of cis-acting factors in the case of the dual reporter constructs (absence of EJC, responsiveness to aminoglycosides, mutational analysis). The RTP calculation in our study is not based on “aminoglycoside-promoted readthrough efficiency”. We only used “basal” readthrough efficiency, which is non-induced readthrough by definition. Also the BTR levels measured by us for inclusion in the iterative model are non-induced (but inducible!), so they were recorded in the absence of readthrough-inducing drugs, indicating that they are entirely dependent on the SCC. While the inducibility of the signal in the assay sufficiently proves that it is real readthrough (because e.g. splicing is not induced by aminoglycosides), we cannot exclude that some of the genes with high BTR are additionally influenced in translational efficiency by introns/EJC. For the case of LDHBx, however, this is unlikely, due to the quantitative correspondence of LDH activity in the peroxisome with readthrough propensity and its very short extension.

We do not think that focusing on BTR is a shortcoming of our approach. The genome-wide screens by [11] and [10] do not identify LDH and none (Eswarappa) or only 1/42 (Dunn) of the other candidates that we have identified. This is also not a shortcoming of their work. It just means that so far no experimental or *in silico* approach is capable of identifying all cases of translational readthrough in mammals. It also indicates that the detailed molecular mechanisms underlying PTR and BTP are likely different. We believe that a combination of our new approach with ribosome profiling and analysis of conservation of ‘non-coding’ regions together with mass spectrometry will identify the ‘extensome’ that is the full set of proteins with above-average readthrough in the proteome.

*3) The authors make a strong point of being the “first” to observe that nonsense suppression allows two mammalian proteins to be derived from the same mRNA. However, this point is also apparent from the supplementary data presented in Welch et al. (2007) (Nature 447: 87-91). Again, this should be acknowledged*.

Thank you for pointing this out! We have now deleted a somewhat misleading line from the Abstract and the beginning of the Discussion. We do not wish to claim, we are ‘the “first” to observe that nonsense suppression allows two mammalian proteins to be derived from the same mRNA’. Throughout the manuscript, we are quoting several papers together reporting more than 50 proteins that have shown this before [[10], [27], [18], and now additionally [9], [11], and [36]]. We have now included a diagram (Figure 2—figure supplement 2) that gives an overview on the experimentally confirmed mammalian readthrough genes. Together, with [11] our study is the first reporting functional translational readthrough in humans, defined as a readthrough event, in which the normal and the extended form have distinct physiological roles and/or localizations. Nonetheless, we have toned down all reference to being “first” in the manuscript.

In our study, we have used the pharmacological induction of readthrough only to support the idea that what we are measuring is indeed readthrough (as opposed to splicing, or RNA editing, see below). We have not included the genes identified by Welch et al. because in our study we only focus on genes undergoing detectable levels of readthrough without drug treatment.

*4) The authors do not discuss/compare the consensus sequence identified from their analyses with any of the stop codon sequence contexts that have been reported in the literature. A comparison with known sequence contexts would be helpful for the readers and may exemplify the strength of the RTP predictor*.

We have now expanded the Discussion section and compare the SCCs with already published readthrough motifs to point out the strength of the RTP predictor. In addition, we have now included a diagram (Figure 2—figure supplement 2) that displays the experimentally confirmed mammalian readthrough genes in context.

*5) The authors use one specific case study, the peroxisomal targeting sequence, to show physiological functions of a readthrough protein. However, adding one more example, other than the PTS, would greatly enhance the applicability of the RTP predictor combined with other in silico analyses. Experimental validation of another case study might be beyond the scope of the paper, but it would show a wider applicability of the method described*.

To enhance the applicability of the RTP algorithm, we now include more *in silico* analysis of the readthrough extensions, containing potential endoplasmic reticulum retention signals, glycosylation sites, transmembrane domains, and possible farnesylation sites (Database 1).

*6) In the experimental validation of the role of readthrough in the synthesis of LDHB+P, the authors should demonstrate the validity of their RTP predictor by doing mutational analyses of the stop codon context according to their consensus models*.

We already were working on an extended mutational analysis of the LDHB+P (now LDHBx) stop codon context according to our consensus model while the paper was under review. In the revised manuscript, we show in Figure 3 the result of our analysis. In addition we have measured more candidates with the LINfs3 consensus (Figure 2).

*7) Although the authors have an intriguing model for the co-import of LDHA with LDHB, It would be important to show two-hybrid data to support it, or address it biochemically using co-IP analyses of the LDHB+P and LDHA*.

We use a two-hybrid (albeit not a yeast two-hybrid) experiment that shows the interaction of LDHBx with LDHA. Numerous descriptions in the literature indicate that any protein can enter peroxisomes when it interacts with a protein that is imported into peroxisomes [eg. McNew et al. TiBS 21, 54-58 (1996), Thoms et al. J Struct Biol 175, 362-371 (2011)]. The two-hybrid assay is based on this co-import and uses LDHA fused with the yellow fluorescent protein (YFP) in combination with LDHBx fused to the C-terminus of the cyan fluorescent protein (CFP). We show that overexpressed YFP-LDHA can only enter the peroxisome, when CFP-LDHBx is co-expressed (Figure 10). When we mutate the targeting signal of LDHBx in the extension, LDHA cannot enter the peroxisome any more (Figure 10—figure supplement 1). We checked if the amino acid extension present in LDHBx could potentially interfere with the interaction of LDHB with LDHA. To answer this question, we now show the structure of tetrameric LDH in Figure 10—figure supplement 2. The C-termini of LDHB are at the very surface of LDH with maximum distance from the interaction surface of the protomers. We can conclude the following from this analysis: (1) The (extended) C-terminus of LDHBx is unlikely to interfere with the oligomerization in the tetramer, and (2) the C-terminal extension is easily accessible to the peroxisomal import receptor PEX5 that must bind the short PTS1 contained in the additional amino acids. If the extension was buried inside the protein, it could affect conformation of LDHBx to interfere with LDHA binding, and the extension would be inaccessible to PEX5, and thereby LDHBx-containing oligomers could not be targeted to the peroxisome.

8) Although they clearly demonstrate that the extended protein is addressed to the peroxysome it is still not clear what is the function of this protein in this organelle. There is no clear demonstration of a physiological function for this protein. Is there any genetic disease linked either to the absence of the PTS1 motif in LDHB or to the modification of the readthrough consensus motif identified by the authors?

At least six previous studies found LDH activity and/or LDH protein(s) in the peroxisome (41; 45, 56; 2, 40; 21). One of them (2) experimentally addresses the function of the protein in this organelle and suggested a role in NAD^+^ regeneration. In spite of all these reports of peroxisomal LDH association, it has never been universally accepted that LDH indeed enters the peroxisome and indeed exerts a function in peroxisomes, because a targeting signal or targeting mechanism could not be identified. In the second, cell biological part of our paper we therefore chose to study the readthrough-dependent targeting (as a function of the readthrough-extension) of LDHBx to peroxisomes. We not only identify a (cryptic) peroxisomal targeting signal, and peroxisomal localization of a protein, but also provide evidence for readthrough-dependent targeting to the peroxisome. We show that targeting (1) occurs in untransfected cells; (2) is dependent on the stop codon: UAA, a tighter stop codon than UGA, abrogates targeting, whereas UGG, a sense-mutation of the stop, improved targeting; and (3) that aminoglycoside –treatment increases the amount of LDH in the peroxisome. Our work provides an answer to this long-standing question, of how LDH enters the peroxisome.

At the moment, the LDHBx extension is expected to be in the 3’UTR, so it is not under scrutiny when it comes to the identification of genetic diseases by exon sequencing. We are however, discussing what is probably the only known disease associated with the LINfs3 consensus (46).

*9) Concerning the molecular mechanism involved. The fact that this readthrough is UGA specific is very intriguing. Indeed up to now the context is known to modify readthrough efficiency but not in such extend. We would expect variations but not a total absence of readthrough product with the two other stop codons. This is reminiscent to the insertion of Sel that is also specific of the UGA codon. It seems that this possibility can be excluded due to the absence of a SECIS element, but can you clearly exclude an alternative mechanism such an editing or an alternative splicing? Supplementary data answering these questions will be important to provide*.

We now show in Figure 3 that the readthrough for LDHBx is indeed UGA specific, because we find absence of readthrough (or only background levels) with the other stop codons or mutations in the LINfs3 consensus. We agree about the absence of SECIS element. However, we analyzed LDHB and the other experimentally tested transcripts regarding potential A-to-I editing sites using the RADAR database (http://rnaedit.com), and based on these results we can exclude RNA editing in these transcripts. We are certain that alternative splicing does not occur that close to the stop codon, because alternatively spliced transcripts would be included in the Ensembl database and as such would have been subject of our genome-wide *in silico* screen. Perhaps the most convincing argument ruling out RNA editing or splicing is our finding that the apparent readthrough is strongly induced (in some cases more the 40-fold) by aminoglycosides in the SCC as well as in the full-length experiments. Neither RNA editing nor splicing are known to be stimulated by these drugs.

*10) It is unclear whether RTP identifies true readthrough candidates, because there are no obvious negative controls to determine whether the readthrough level is “programmed”*.

*The authors select fifteen stop codon contexts scored by the lin model, and measure their readthrough rates using an eYFP-stop-luciferase reporter. The observed readthrough rates* – *with the exceptions of MDH1 (2.91% readthrough) and LDHB (1.55%) are quite low, ranging from 0.1-0.66%. This is a much lower amount of readthrough than is observed for bovine VEGF-Ax (10%,*
[11]*) or various human genes (ranging from 0.7 to 60%,*
[10]*). It is unclear whether this amount of readthrough represents a basal level of readthrough that could be obtained with more or less any stop codon context, or an elevated level of readthrough due to something special about the nucleotide contexts selected by the lin model*.

*To address this, the authors should include in this assay a set of negative controls, including a construct lacking luciferase, and a set of contexts with very low RTP scores, to establish a background readthrough rate. The authors should then discuss how far above background, if at all, their readthrough candidates are*.

Our data and experiments contain negative controls in the sense that we have included many genes with low RTP and therefore a low BTR, for example PPP1R3F (0.18%), or PRDM10 (0.13%), or THG1L (0.15%) (Figure 1 and Table 1). Following the reviewers’ suggestion, we have now included a new negative control that is even better suited than a luciferase-less control to establish the background levels: We use a construct that has two successive stop codons. This construct shows a BTR of 0.13%, indicating that this is the background level below we cannot distinguish between readthrough and experimental noise. As we neither want to obscure this fact, nor do we want to artificially scale our data, we decided not to subtract this value in the BTR measurement. Instead we indicate the background level in Figure 1 by a red line. As stated above, due to the specific form of the dual reporter assay we are using, cis-element dependent readthrough cannot be the origin of the readthrough levels we are measuring. The high-RTP consensus emerged from data that was in no way primed to readthrough, so we are confident (even if we cannot prove it), that readthrough of more than 5%, depending on the cell type, cannot be obtained by BTR. For PTR ([11], and probably also Dunn et al., even though the latter study does not analyze the mechanism leading to readthrough) there seem to be no upper limit of readthrough levels, depending on the type of cis-elements and trans-acting factors, even 100% readthrough seem possible. Again, we are very thankful that the reviewers’ comments have given us the opportunity to distinguish between cis-element dependent readthrough (PTR) and SCC-dependent readthrough (BTR). It is clear that BTR can (Loughran et al.) but need not (Eswarappa et al.) be modulated by PTR. And, as stated above and below, the BTR of approx. 1.5% for LDHBx is not low, because, as we explain in the Discussion, it leads to a roughly equal distribution of LDH in the peroxisome and in the cytosol. With higher readthrough, the peroxisomal LDH activity would exceed the cytosolic! Regarding our prediction of readthrough for all human transcripts we are aware that many of the listed RTP values correspond to what could be interpreted as the ‘baseline’ of measurable BTR. We are also aware that negative RTP values are somehow counter-intuitive, as they would suggest a negative BTR, which is not possible. Here, a calibration of RTP values to BRT probabilities by means of, e.g., analysis of the distribution of RTP scores could help to assess more intuitively interpretable RTP values. However, with the limited dataset and the apparent nonlinear dependence of BTR on the SCC a calibration is not possible at the moment. A workaround would be to just include the first 144 (consensus) sequences in Dataset 1 (or trim the list according to the first negative control or an arbitrary threshold), but we feel that readers could be interested in the complete list. (Another possibility for avoiding negative RTP would have been to use “constrained” estimation methods, i.e. methods that avoid negative coefficients. However, for our first approach we wanted to use the simplest methods possible to take into account the limited number of sequences.) We hope that extended datasets of BTR measurements and nonlinear methods in the future will provide improved RTP values.

*11) The authors claim that RTP is a genome-wide predictor of readthrough, but they do not substantiate this claim with genome-wide data. The authors should provide a genome-wide examination of readthrough using some sort of experimental measure. For example, they could estimate readthrough rates genome-wide using published ribosome profiling data, and compare these to the predictions made by RTP*.

We have now linked our genome-wide *in silico* screen better to existing experimental data on translational readthrough. We have analyzed more candidates to validate the LINfs3 (Figure 2). We have extended the Discussion on the hits found in several studies comparing them in more detail. We have also added a figure (supplement to Figure 2) describing the overlap between our data and the previously identified readthrough proteins. This includes a search for existing readthrough proteins for the consensus described in our paper and an extended the discussion on why there is relatively little overlap between the different approaches applied so far. And we have extended the discussion on the regression between measured BTR and RTP.

*12) The authors claim that RTP is quantitative, but provide no analysis of its quantitative performance. The authors should provide an explicit analysis demonstrating that RTP is quantitative, or a discussion of why it might not be, especially in light of the fact that they acknowledge that “...there is non-linear contribution to RTP in the sense that other high-RTP genes are likely to show readthrough levels higher than predicted.” Such an analysis might be a quantitative comparison (e.g. a scatter plot and a Pearson correlation coefficient) of predicted and observed readthrough rates for all stop codon contexts tested, or an ROC curve describing the sensitivity and specificity of RTP at different score thresholds. Finally, because RTP is a new metric, descriptive statistics on the RTP score distribution should be provided to facilitate interpretation of individual scores*.

We added scatter plots showing the correlation of BTR and RTP for 81 sequences for the LINiter as well as the LINfs3 model (Figure supplements to Figures 1 and 2). Here, the RTP values have been obtained from predictions within the leave-one-out cross-validation and are therefore an indicator of the generalization capacity of our approach. The plots show a nonlinear dependency of BTR and RTP for both models, indicating that nonlinear regression models could provide a better fit. The Pearson correlation for both models is significant but weak with the LINfs3 model showing a slightly higher correlation (0.41, p=0.0001) than the LINiter model (0.34, p=0.0022). This suggests that the LINfs3 model with its only 15 parameters is more suitable for our purpose than the LINiter model with 51 parameters. Due to the lack of true negative readthrough (readthrough 0%) examples a computation of classification performance indices such as sensitivity and specificity is not possible. We do not claim RTP to be a new metric; we rather introduce this notion to keep the formulation throughout the manuscript as concise as possible.

*13) As mentioned above, the readthrough rates observed for the vast majority of candidates identified in this work are very low compared to known examples. It therefore seems likely that stop codons undergoing higher levels of readthrough were either missed by RTP, or not tested by the experimenters. In addition, certain cases of readthrough are regulated in rodents (*[59]*) and flies (*[10]*; Robinson & Cooley, 1997) or differ between different wild-type yeast strains (Torabi & Kruglyak, 2011). Consistent with these observations, the authors themselves note that LDHB undergoes far more readthrough in human U118 cells (∼5%) compared to human fibroblasts and HEK cells (1-2%). It is therefore possible that these models could be overfit to the cell type in which the experiment was performed, in addition to the specific contexts on which the models were trained. In light of these facts, the authors must establish the scope of their claims*.

The BTR readthrough rates are in the low percent range. As detailed above, we provide evidence that these rates are entirely dependent on the SCC and do not require cis-elements or trans-factors. Considering our finding that readthrough of some of the LINfs3 consensus candidates are at least 10-fold above the physiological background, we do not think that, from the perspective of a stop codon and its’ usual function to stop translation, these are low rates. It is clear that the RTP approach would miss any cases of readthrough that is independent of the stop codon, and we now state this more clearly in the Introduction, the Results and in the Discussion. Overfitting: we are aware that the number of 66/81 sequences represents the lower limit for a multivariate regression on 51 dimensions. However, with that knowledge in mind we deliberately restricted our analysis to linear models, applied a rigorous model validation using a leave-one-out strategy and performed a feature selection procedure to further reduce the number of model parameters (to 15 in the case of LINfs3). The fact that we could identify the relevant position of the SCC as well as the consensus motif for high RTP assured us that our model generalizes as good as possible for linear approaches. Furthermore, we pave the way for future extended datasets that now can be built by concentrating on the relevant SCC positions. This reduces the number of experiments for fully specified BTR assessment based on the SCC considerably (from 4^12*3 ∼ 50,000,000 to 4^3*3=192) and will allow the application of nonlinear models for RTP prediction. The additional experiments we have added to test the LINfs3 consensus (Figure 2) suggest that the predictions derived from one cell type in principle can be applied to another, although the overall level of BTR may differ between cell types.